# Improving old tricks as new: Young adults learn from repeating everyday activities

Gil Meir Leizerowitz[1,2]*, Ran Gabai[3], Meir Plotnik[4,5], Ofer Keren[2,6], Avi Karni[1,7,8]*

1 Sagol Department of Neurobiology, University of Haifa, Haifa, Israel, 2 The Rehabilitation Hospital, C. Sheba Medical Center, Ramat Gan, Israel, 3 Technion–Israel Institute of Technology, Haifa, Israel, 4 Center of Advanced Technologies in Rehabilitation, C. Sheba Medical Center, Ramat Gan, Israel, 5 Faculty of Medicine & Sagol School of Neuroscience, Department of Physiology and Pharmacology, Tel Aviv University, Tel Aviv, Israel, 6 Galilee Rehabilitation Center, Karmiel, Israel, 7 The E. J. Safra Brain Research Center for the Study of Learning Disabilities, University of Haifa, Haifa, Israel, 8 Department of Diagnostic Imaging, C. Sheba Medical Center, Ramat Gan, Israel

* Gileizer@gmail.com (GML); avi.karni@yahoo.com (AK)

**Data Availability Statement:** The anonymized data base have been made public in Open Science Framework (https://osf.io/) - DOI 10.17605/OSF.IO/XVSQ4. In addition, our data were uploaded as a Supporting Information files.

## Abstract

The notion that young healthy adults can substantially improve in activities that are part of their daily routine is often overlooked because it is assumed that such activities have come to be fully mastered. We followed, in young healthy adults, the effects of repeated executions of the Timed-Up-and-Go (TUG) task, a clinical test that assesses the ability to execute motor activities relevant to daily function—rising from a seated position, walking, turning and returning to a seated position. The participants (N = 15) performed 18 consecutive trials of the TUG in one session, and were retested on the following day and a week later. The participants were video recorded and wore inertial measurement units. Task execution times improved robustly; performance was well fitted by a power function, with large gains at the beginning of the session and nearing plateau in later trials, as one would expect in the learning of a novel task. Moreover, these gains were well retained overnight and a week later, with further gains accruing in the subsequent test-sessions. Significant intra-session and inter-session changes occurred in step kinematics as well; some aspects underwent inter-sessions recalibrations, but other aspects showed delayed inter-session changes, suggesting post-practice memory consolidation processes. Even common everyday tasks can be improved upon by practice; a small number of consecutive task repetitions can trigger lasting gains in young healthy individuals performing highly practiced routine tasks. This new learning in highly familiar tasks proceeded in a time-course characteristic of the acquisition of novel 'how to' (procedural) knowledge.

## Introduction

The possibility that adults can continue to improve, i.e., undergo additional learning, in tasks that constitute their daily routine has not been studied, mainly because of the commonsense notion that participants are, by adulthood, highly trained in such tasks. It is tacitly assumed, therefore, that even with additional practice, significant and enduring gains in the

**Funding:** The study was supported in part by the Edmond J. Safra, Brain research center, University of Haifa (grant 100007648/18 to A.K). The funders had no role in the design of the study, data collection and analysis, decision to publish, or preparation of the manuscript.

**Competing interests:** The authors have declared that no competing interests exist.

performance of activities that are part of adults' everyday routine would be undetectable [1,2]. The aim of the current study was to test whether a set of common daily routines, specifically those included in the Timed-Up-and-GO test (TUG), a clinical test of daily function, can be improved upon through practice in young healthy adults, and whether such gains can be retained. The study was designed to address the possibility that if gains in performance are achieved after practicing common daily activities, such as those comprising the TUG, these gains will evolve with a time-course similar to that of acquiring skill in simple tasks practiced for the first time in laboratory settings e.g., [3–5].

The learning of task-related movement routines and, specifically, the generation of long-term procedural memory for the performance of movement sequences, are often characterized by distinct phases which have been delineated in a number of laboratory tasks e.g., [4–14]. In these laboratory tasks, rapid gains in performance occur early on in training ("fast learning"), but after a certain number of within-session task iterations, if the task demands remain unchanged, performance levels off e.g., [8,11,15–17]. Both these phases, in the acquisition of new skills, are well modeled by power functions [8,11,15,18,19]. Studies have shown however that the 'plateau phase' can be followed by a latent, post-session, phase wherein the knowledge gained in practice becomes consolidated as long-term gains, with stabilization and reduced susceptibility to interference from competing or conflicting experiences, or even lead to additional gains in performance (delayed 'offline' gains) in the hours that follow the termination of training e.g., [11,20]. In some tasks, in adults, an interval of sleep is required before delayed gains can be expressed [4,10,21–23]. Stabilization and delayed 'offline' gains in performance are considered to reflect procedural memory consolidation processes, i.e., generative processes that lead to the establishment of improved, stable, task solution routines that can be retrieved and even further honed over long-time intervals [24–26].

The different phases in motor skill acquisition have been studied mostly in the context of finger and hand movements, for example, the finger opposition sequence (FOS) learning task e.g., [27] or its variation, the finger tapping task e.g., [20], in visuomotor tasks [16,28] or in more complex manual tasks e.g., [26,29]. These phases were also delineated in the serial reaction time task (SRT) wherein participants are not aware of the stimuli-movements sequence they should be mastering e.g., [5].

The time-course of performance changes in more common everyday motor routines, such as walking or upright standing, have been less studied [8,30,31] and when studied, these tasks were followed in special laboratory conditions, i.e., not in everyday or well-rehearsed conditions. The generalizability of the results from studies using laboratory tasks to the learning of real world motor skills has been questioned [32,33]. Some evidence, however, does suggest that there is a correspondence between the time-course of learning and skill consolidation in the acquisition of voluntary manual skills (e.g., FOS or finger tapping) and the time-course of learning and skill consolidation in more gross motor tasks e.g., balance maintenance tasks, [8,18,34]. Also, as in the FOS learning task [10,23], sleep can facilitate off-line gains in tasks such as walking an irregular elliptical path while doing mental arithmetic [30]; although the type of task and training parameters may moderate the necessity of sleep [35].

Because the characteristic phases in the mastering of new tasks can be delineated in tasks other than motor skills (e.g., perceptual skills [17]) it has been proposed that the acquisition of a new skill (i.e., the generation of long-lasting practice-dependent gains in ability) reflects a common repertoire of basic neural processes, albeit in the different brain systems that subserve the execution of the different tasks e.g., [4,24]. Thus, the time-course of skill acquisition rather than the specifics of the task may best reflect processes of neural (as well as systems level, brain plasticity [36]). This would suggest that if young adults do acquire new or additional skill when afforded the opportunity to practice common everyday activities, the time-course of

performance improvement would follow that of new skill learning in laboratory tasks and conditions. However, we do not expect that the characteristic phases of new learning will replicate in a task which is already over-trained, as in walking or typing. In other words, at issue is the question of whether we should expect to see the characteristic time-course of new skill acquisition (rather than fast adaptation [33,37–39], and effective transfer [40] occurring in well-rehearsed (familiar) tasks such as, for example, typing when we introduce a new text or when rising from an unfamiliar chair. Viewed from a related framework there is the notion that very extensive, long-term and deliberate practice is required to gain, albeit via incremental gains, in order to attain expertise in well rehearsed tasks in different domains [41]. Thus the question would be whether, if afforded an opportunity for deliberate practice, young adults would show a pattern of continued incremental improvement or whether a deliberate-structured practice opportunity would result in initiating a form of 'de novo' learning for a highly practiced, everyday task [33,39,42,43].

It has been suggested that different neuronal mechanisms underlie the recalibration of an existing skill or sets of skills, versus the acquisition of a new task performance controller mechanism and the generation of a new performance routine ('de novo' learning) for the trained task [24,33]. A fast learning phase in the initial learning session [11,17], a between-sessions shift in speed-accuracy tradeoff [11,38] and subsequently long-term, "offline", gains between sessions [11,17,38] have been suggested as characteristic of the 'de novo' building of a task specific controller and the generation of a new performance routine.

The task we used to test this prediction was the TUG test, because it consists of a sequence of activities which are repeatedly executed on a daily basis [44]. The TUG test is often used to assess mobility and risk of falling and to evaluate the progression of individuals engaged in rehabilitation programs [44,45]. The reliability of the TUG test was found good and even excellent (ICC 0.75–0.99) in multiple studies, which suggests that little learning occurs on repeating the task e.g., [46,47]. In many studies, the notion that learning or adaptation (performance improvement with task repetition) may occur in the TUG was considered so unlikely that TUG results were computed as the average of two or three successive trials e.g., [48]. However, several studies have indicated that there might be an effect of accumulating experience (retests), with performance improving when the task is re-iterated [49–52]. For example, in community-dwelling elders, TUG assessments over a period of several months showed that the TUG test completion times was significantly shorter in later retests [52,53] and the correlation for two TUG tests separated by intervals of a few months was moderate (ICC = 0.56) [54]; lasting practice related effects were not considered in explaining these results. Improvements in TUG test scores, on successive trials, in the context of recuperation and rehabilitation, across a one week and longer interval, was ascribed to rapid familiarization with the test, i.e. a learning effect [49–51]. Thus, there are indications for a possible process of learning in the TUG paradigm.

The aim of the current study was to test whether the performance of common daily activities, included in the TUG test, can be significantly altered (improved) through structured task repetition in young healthy adults; specifically, whether improvements in TUG performance will show fast adaptation or effective transfer of the existing well-honed sub-skills to the particular requirements of the training session, or rather suffice to induce de novo learning, 'fast learning' and plateau and subsequently post-session consolidation phase changes in performance and kinematics. Our work hypothesis was that affording young healthy adults an opportunity to practice (repeat in a deliberate-structured manner) a sequence of common, everyday, highly trained sub-skills, may suffice for initiating 'de novo' skill learning. We show that TUG practice induced robust and enduring gains in performance as well as robust and

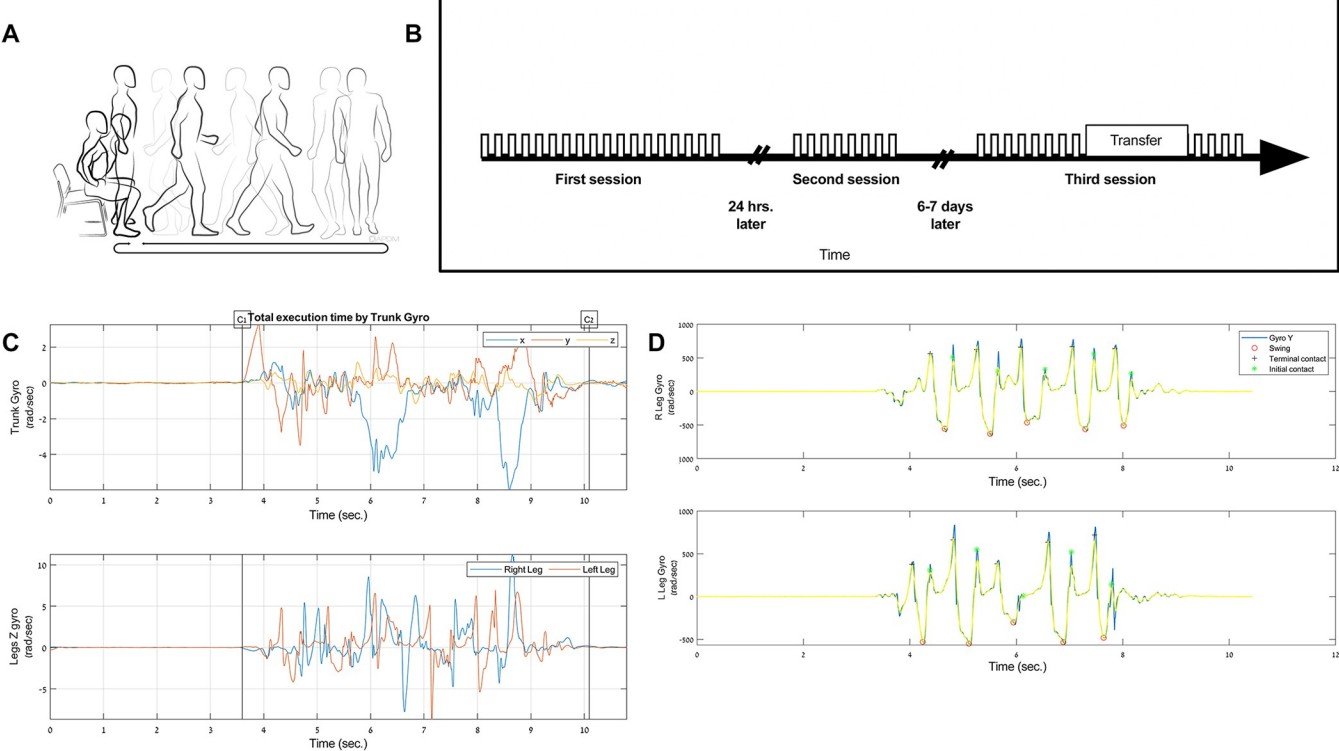

**Fig 1. Study design and performance measurements.** (A) An illustration of the TUG task (WWW.APDM.COM). (B) The three study sessions. In the transfer test (seesion-3) the TUG was performed in the reversed direction (results not shown). (C) Horizontal lines indicate the beginning and end of the trial; the cutoffs are shown for the trunk IMU's gyroscope trace (blue–X axis; red–Y axis; yellow–Z axis) (upper panel) and for the leg IMUs gyroscope (z-axis; blue–right leg, red–left leg) (lower panel). (D) Gyroscope data from both legs in the y-axis used for identifying steps in the walking segment of the TUG (upper panel–right leg; lower panel–left leg).

enduring changes in the kinematics of the everyday activities when performed in the context of the practiced task.

To test whether TUG task repetition (practice) leads to new learning and, importantly, to the retention of the gains, and whether such learning occurs in a manner similar to the acquisition of skill in novel, unfamiliar, tasks, we followed 15 young healthy adults in three sessions. Fig 1 shows the TUG task (A) and the overall plan of the three study sessions (B): an initial practice session (session-1), an overnight re-test (session-2) and a final re-test six to seven days later to assess long-term retention (session-3). The time to complete each TUG iteration (TUG duration) was derived, independently, from two sources: *i*. the video recording of the trial, and *ii*. the time-series data from the gyroscope in the participant's chest IMU (Fig 1C) (Materials and methods). The initial and terminal contact points of every step from the leg IMU gyroscopes (Fig 1D) were used to parse the accelerometers' data obtained from the trunk and lumbar (see S1 Fig) and determine mean, maximal and minimal acceleration for each step in the walking segment of the TUG, and subsequently compute the respective mean step values for each TUG trial.

## Results

### Practice changed TUG performance during session-1

Overall, TUG duration was significantly shortened across the first session (trial 1 to trial 18) in all participants; by 15.0%±5.3 (range: 4.8% - 25.7%) according to the video recordings and by

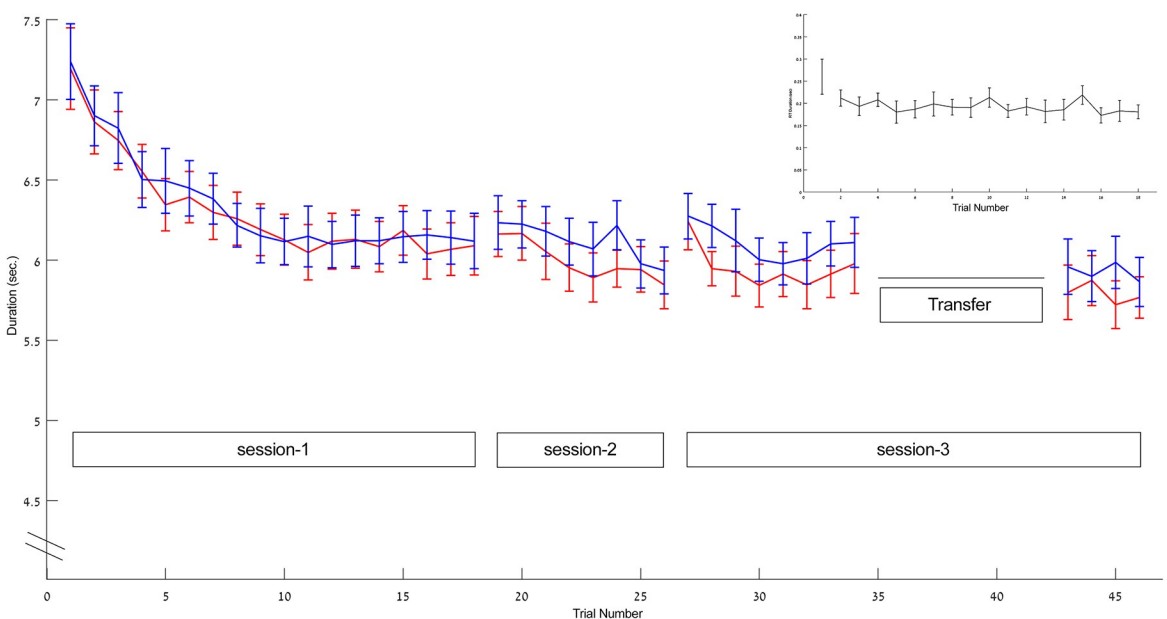

**Fig 2. Trial duration in the three different sessions according to both methods of measurements.** Red–data derived from video analysis, Blue–data derived from IMUs. The gains accrued in session-1 were on the order of 1.11±0.51 sec. (video). Inset–time from the 'go signal' to first postural adjustment (RT). Data are represented as mean ± SEM. (For the corresponding bar graph depiction of the average performance changes within and between segments/sessions see S2 Fig).

15.0%±7.7 (range: 2.5% - 35.5%) according to the trunk IMU. A comparison of the first segment (4 trials) of the session to the final segment (4 trials) again showed the significant improvement in TUG duration ($F_{(1,12)} = 68.68$, $p<0.001$, $_p^2 = 0.851$; $F_{(1,12)} = 27.39$, $p<0.001$, $_p^2 = 0.695$; video and IMUs, respectively) but also a significant interaction ($F_{(3,39)} = 5.907$, $p = 0.008$, $_p^2 = 0.282$; $F_{(3,39)} = 8.689$, $p<0.001$, $_p^2 = 0.401$; respectively) indicating that the rate of change across trials decreased significantly across the session (Fig 2). As can be seen in Fig 2 the video and IMUs derived measurements of TUG duration were highly correlated across the three sessions ($r_p = 0.858$, $p<0.001$, N = 678) and there was no significant difference between the two measures across the 18 trials of the session ($F_{(1,12)} = 0.275$, $p = 0.61$, $_p^2 = 0.022$).

The gains in TUG duration attained across the session were not due to a shortening of the time to initiate the TUG movements sequence after the 'go signal', i.e., reaction time (RT, video based) (Fig 2–inset). In session-1, the overall gains in RT were very small 0.08±0.18 seconds (sec.); there was a small but not significant gain in RT between the first and second TUG trials of the session (paired t-test, $t_{14} = 1.61$, $p = 0.13$, Cohen's d = 0.12) and RTs in trials 2–5 did not differ from RTs in trials 15–18 ($F_{(1,12)} = 1.696$, $p = 0.217$, $_p^2 = 0.124$). Moreover, the group mean linear trend line from the 2nd to the 18th trial was y = -0.001x+0.2013; the slopes of the individually fitted linear trend lines, generated for each of the participants, were not significantly different from zero ($t_{14} = -1.252$, $p = 0.231$, Cohen's d = 0.003).

Overall, the gains in group mean TUG duration across session-1 were highly-correlated with a power function model ($R^2 = 0.95$, 0.94, video and IMUs, respectively) (Fig 3A). However, two distinct phases of learning were clearly identifiable; an initial phase in which TUG duration became increasingly shorter on consecutive trials (trials 1–9, fast learning), and a second phase wherein performance remained quite stable (trials 10–18, plateau). Trials 1–9 of session-1 were well-fitted by a power function ($R^2 = 0.98$, 0.97, video and IMUs, respectively) while a linear trend with a very minimal slope fitted trials 10–18 (y = -0.0033x+6.1154,

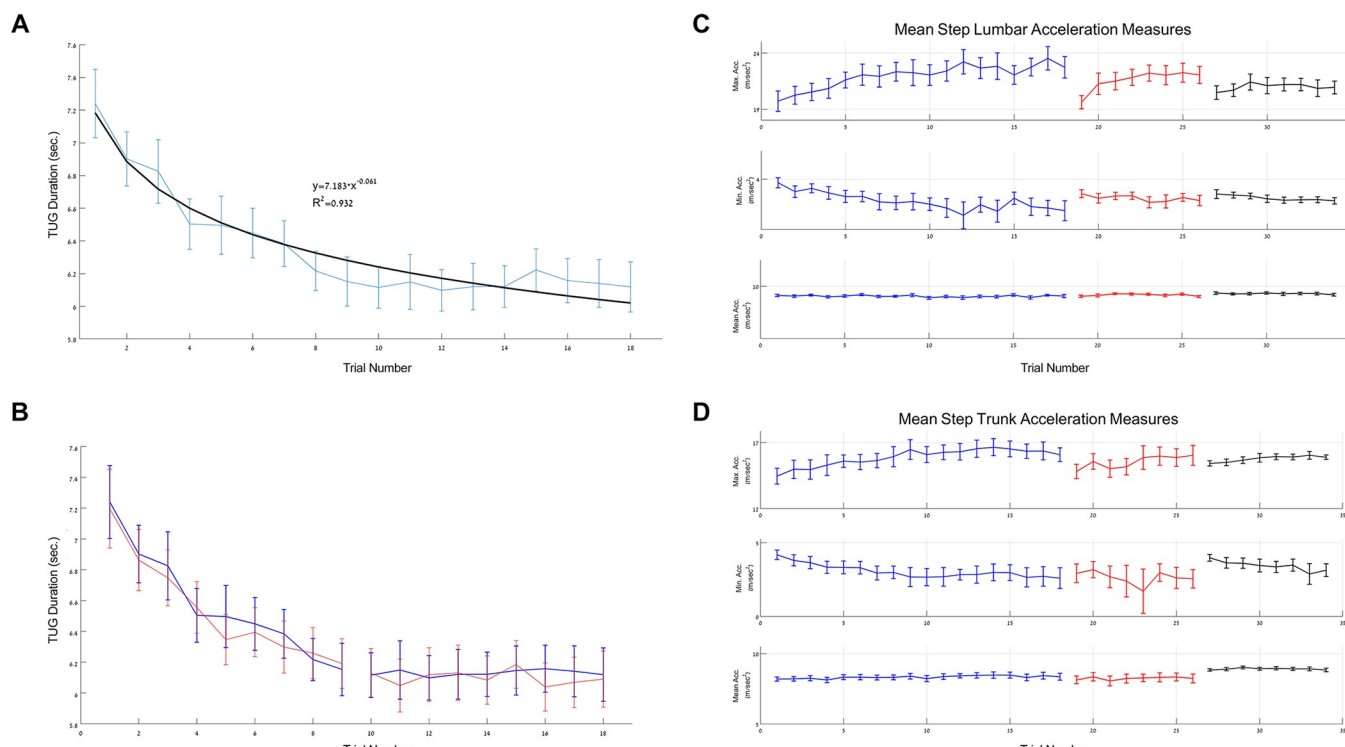

**Fig 3. The time course of TUG learning.** (A) Group mean plot of the TUG duration in successive trials of session-1, according to the trunk gyroscope (blue), and a power function model fitted to the data (black, inset). (B) Group mean plot of the TUG duration in the two phases of session-1 (Trunk gyroscope (blue) and video (red). (C, D) Mean-step acceleration measures across TUG trials in sessions 1–3; lumbar and trunk IMUs (respectively). Upper panels—maximal acceleration; middle panels—minimal acceleration; lower panels—mean acceleration. The mean-step maximal and minimal acceleration measurements were well-fitted by a power function model; lumbar ($R^2$ = 0.89, $R^2$ = 0.76, respectively) and trunk ($R^2$ = 0.89, $R^2$ = 0.90, respectively) IMUs. Data are represented as mean ± SEM.

y = 0.0022x+6.118 video and IMUs, respectively; a one-sample t-test showed that the trend line slopes at the latter part of session-1 were not significantly different from zero ($t_{14}$ = -0.371, p = 0.716, Choen's d = 0.037; $t_{14}$ = 0.183, p = 0.858, Choen's d = 0.05, video and IMUs, respectively) (Fig 3B). A repeated measures ANOVA comparing trials 1–9 to trials 10–18, showed a significant difference between the two phases ($F_{(1,11)}$ = 69.04, p<0.001, $_p^2$ = 0.863; $F_{(1,11)}$ = 32.519, p<0.001, $_p^2$ = 0.747, video and IMU based TUG duration, respectively), a significant Trial effect ($F_{(8,88)}$ = 12.06, p<0.001, $_p^2$ = 0.52; $F_{(8,88)}$ = 8.77, p<0.001, $_p^2$ = 0.444) as well as a significant Phase X Trial interaction ($F_{(8,88)}$ = 11.24, p<0.001, $_p^2$ = 0.506; $F_{(8,88)}$ = 11.23,p <0.001, $_p^2$ = 0.505).

The learning process in session-1 was also reflected in kinematic measures. As can be seen in Fig 3C and 3D, mean-step measures, such as the maximum and minimum acceleration in the x-axis (up-down movements) derived from the trunk and lumbar IMUs during the walking segment of the TUG trials in session-1, reflected a time-course similar to that of the TUG duration gains. The relative improvement in TUG duration during session-1 ([trial1—trial18]/ trial1) was correlated with the corresponding learning-related changes in mean step maximal acceleration according to the lumbar IMU but not with the changes in mean step minimal acceleration (Spearman, $r_s$ = -0.593, p = 0.02; r = 0.154, p = 0.58, respectively). Nevertheless, not all kinematic measures improved as a function of task iteration (mean-step mean acceleration, Fig 3C and 3D). There was no significant difference between males (6/15) and females in terms of the magnitude of the TUG duration gains attained in session-1. (Independent t-tests,

$t_{10.066} = -0.147$, $p = 0.886$, Choen's $d = 0.055$, $t_{13} = -0.637$, $p = 0.535$, Choen's $d = 0.078$, video and IMU based TUG duration, respectively).

## Practice effects were retained across sessions

The gains in TUG duration were well retained overnight (Fig 2). A comparison of TUG duration in the final four trails of session-1 to the initial 4 trials of session-2 showed no significant session effect ($F_{(1,12)} = 0.688$, $p = 0.423$, $_p^2 = 0.054$) nor a trial effect ($F_{(3,36)} = 0.571$, $p = 0.638$, $_p^2 = 0.05$) (IMUs measurements, for the corresponding video measurements see supplementary results; S1 File). The offline, between sessions, differences in TUG duration were -1.5 ± 3.8 percent for the interval between session-1 and session-2. Sleep quality, as assessed by the PSQI questionnaire score, did not correlate with the overnight change in performance (Spearman, $r_s = -0.046$, $p = 0.872$; $r_s = -0.188$, $p = 0.519$; video and IMU based, respectively) nor with the overall gain across the three sessions (Spearman, $r_s = -0.67$, $p = 0.813$; $r_s = 0.-0.083$, $p = 0.768$, video and IMU based, respectively). Furthermore, additional significant gains occurred within the second session, i.e., between trials 1–4 and trials 5–8 ($F_{(1,14)} = 5.796$, $p = 0.03$, $_p^2 = 0.293$).

The gains in TUG duration attained by the end of session-2 were well-retained by session-3, i.e., over a one-week interval ($F_{(1,14)} = 1.727$, $p = 0.210$, $_p^2 = 0.110$). The offline differences in TUG duration from the end of session-2 to the beginning of session-3 were on average -4.0 ± 4.4 percent. Moreover, additional gains in TUG duration were accrued across session-3, with faster performance in trials 5–8 compared to trials 1–4 of the session ($F_{(1,14)} = 6.626$, $p = 0.022$, $_p^2 = 0.321$); TUG duration continued to improve in trials 17–20 compared to trials 5–8 of the session ($F_{(1,14)} = 6.289$, $p = 0.025$, $_p^2 = 0.310$). Thus, overall, the gains in TUG duration across the three sessions, were on average 15.08% (range: 6%-30%) and 18.0% (range: 0%-37.3%) computed to before and after the transfer test segment, respectively.

## Step kinematics between-sessions–intersession recalibration and retention

Although the TUG duration improved throughout the three sessions, several kinematic measures did not show retention from the last four trials of session-1 to the first four trials of the following day. As can be seen in Fig 3C and 3D the mean step maximal acceleration attained by the end of session-1 was significantly reduced at the initial trials of session-2 in the x-axis (up-down direction) ($F_{(1,12) =} 11.60$, $p = 0.005$, $_p^2 = 0.491$, $F_{(1,12)} = 6.01$, $p = 0.03$, $_p^2 = 0.334$, lumbar and trunk IMUs, respectively). In fact, there was no significant difference, in mean step maximal acceleration, between the first four trials of session-1 and the first four trials of session-2 ($F_{(1,14) =} 2.148$, $p = 0.165$, $_p^2 = 0.133$; $F_{(1,14)} = 0.107$, $p = 0.748$, $_p^2 = 0.008$, lumbar and trunk IMUs, respectively) suggesting that a recalibration occurred overnight. Similarly, the practice-related gains in mean step maximal acceleration in the x-axis that were re-attained in session-2 did not affect performance in the initial trials of session-3, one week later; mean step maximal acceleration at the beginning of session-3 did not differ from the acceleration at the beginning of session-1 ($F_{(1,13)} = 0.393$, $p = 0.541$, $_p^2 = 0.029$; $F_{(1,14)} = 1.013$, $p = 0.331$, $_p^2 = 0.068$, lumbar and trunk IMUs, respectively).

Recalibration, however, was not the only pattern of between-session effects. Other aspects of step kinematics showed robust intersession retention, with the learning gains attained in one session expressed in the walking segment of the TUG at the very beginning of a subsequent session. For example, the maximal acceleration in the z axis (forward-backwards), recorded by the lumbar IMU, in the first four trials of session-2 and session-3 were significantly increased compared to the first four trials at the beginning of session-1 ($F_{(1,14)} = 5.85$, $p = 0.03$ $_p^2 = 0.295$; $F_{(1,14)} = 10.139$, $p = 0.007$, $_p^2 = 0.42$ respectively) (Fig 4). Moreover, the

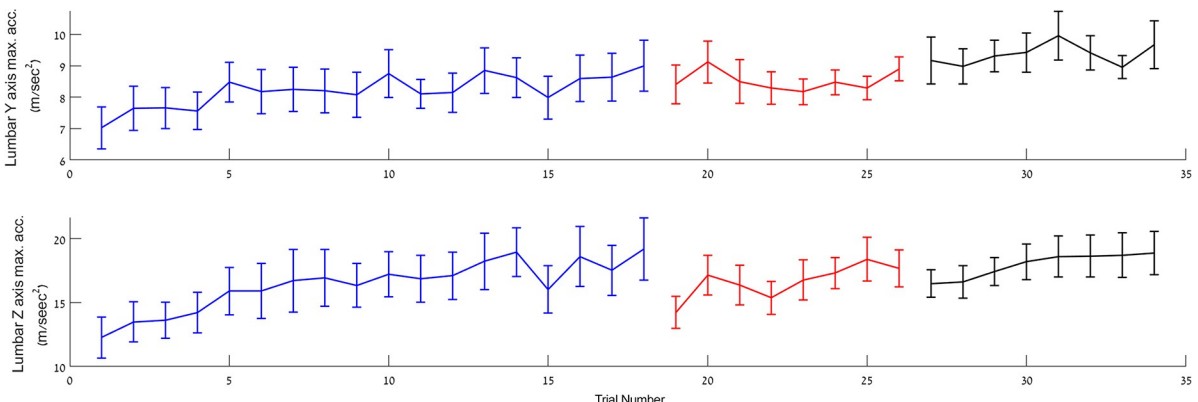

**Fig 4. Mean step maximal acceleration in the walking segment of the TUG recorded by the lumbar IMU.** Upper panel–acceleration in the Y axis (left-right); lower panel–acceleration in the Z axis (forward-backward). Data are represented as mean ± SEM.

gains attained by the last four trials of session-1 were retained in the beginning of session-2 ($F_{(1,12)} = 1.917$, p = 0.191, $\eta_p^2 = 0.138$) and the gains attained in the last four trials of session-2 were retained at the initial four trials of session-3 ($F_{(1,14)} = 0.158$, p = 0.697, $\eta_p^2 = 0.011$).

## Delayed post-practice changes in mean-step mean acceleration

While certain aspects of the kinematic of the walking segment of the TUG have undergone recalibration, between sessions, and other aspects showing clear intersession retention of the practice effects, some mean-step kinematic measures were found to improve between-sessions, specifically, across the one-week inter-session interval between session-2 and session-3. Fig 5C shows that the mean step mean acceleration in the x-direction, measured in the trunk and lumbar IMUs, improved in the interval between the end of session-2 and the beginning of session-3 ($F_{(1,28)} = 4.913$, p = 0.035, $\eta_p^2 = 0.149$, combined trunk & lumbar IMUs). In addition, there was a significant decrease in the individual between-trial variance in mean-step mean acceleration as recorded in the trunk IMU, i.e., participants had reduced variance across the successive TUG trials in in the initial segment of session-3 compared to the level of between-trials variance they exhibited in the final segment of session-2 (paired t test, $t_{(14)} = 2.213$, p = 0.044, Cohen's d = 0.57, within subject comparison) (Fig 5D). The (offline) gains in terms of the mean step mean acceleration in the x-direction, measured in the trunk and lumbar IMUs, (final segment of session-2 and initial segment of session-3) were on the order of a 6% improvement compared to performance at the end of session-2 (final segment).

## Discussion

The current results show that the TUG task, a task composed only of highly familiar sub-tasks and routines of daily activity, can be further improved even in young healthy adults repeating the task in a specific set-up. The repeated execution of everyday activities can initiate a new learning process and, moreover, lead to the consolidation of the gains into long-term 'how to' memory. Robust learning and good retention were manifested in the TUG execution time, as well in kinematic aspects of TUG performance. Learning and significant retention were found, in distinct kinematic parameters, even in the highly-familiar walking segment of the TUG.

Importantly, learning and retention during, and following, TUG practice proceeded in the distinct time-course of learning and retention previously described in the acquisition of new finger movement tasks e.g., [11], as well as when participants practiced gross body movements

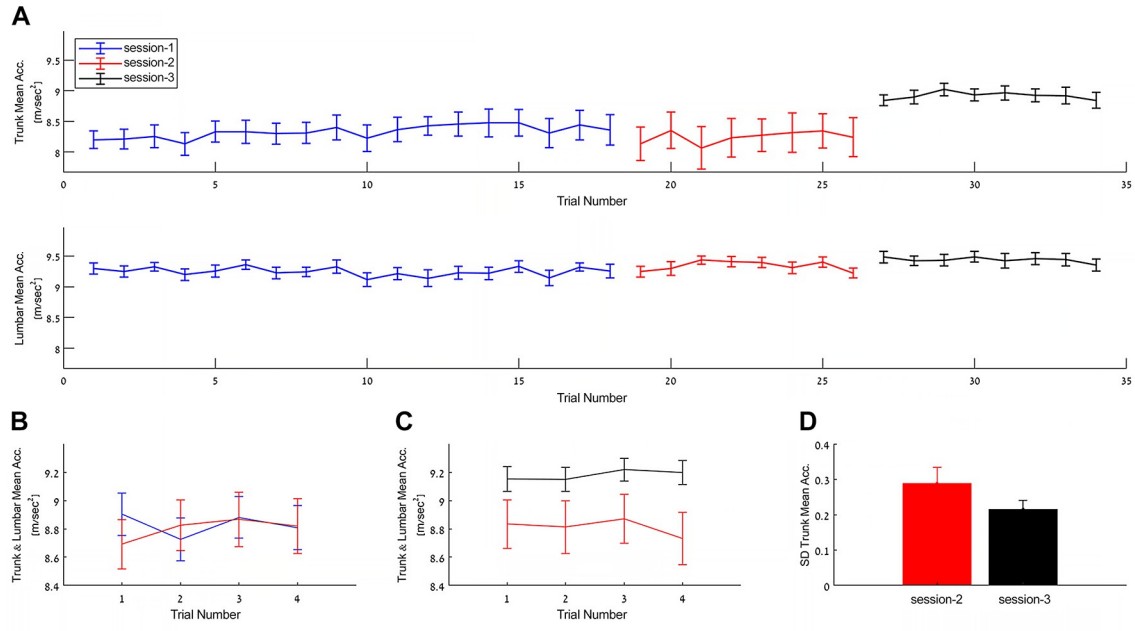

**Fig 5. Mean step mean acceleration in the x-axis (up-down) in the walking segment of the TUG across the three sessions.** (A) Data from the trunk (upper panel) and lumbar (lower panel) IMUs. (B) Combined trunk and lumbar measurements from trials 15–18 of session-1 (blue) compared to trials 1–4 of session-2 (red). (C) Combined trunk and lumbar measurements from trials 5–8 of session-2 (red) compared to trials 1–4 of session-3 (black). (D) Group mean standard deviations of the mean step mean acceleration from the trunk IMU in session-2 (red) compared to session-3 (black). Data are represented as mean ± SEM.

in novel, unfamiliar, laboratory conditions e.g., [8]. Thus, in the well-rehearsed (familiar) sequence of rising-walking-sitting composing the TUG task, the gains followed the characteristics time-course of de novo learning, 'fast learning' and plateau, rather than fast adaptation [33,37] or effective transfer [40]. Moreover there is evidence for a (post-session) consolidation phase in the form of 'delayed' (and well retained) changes in kinematics in the walking segment of the TUG [11,38].

Learning in session-1 clearly followed the two distinct phases, fast learning and plateau phase, that were described in perceptual and simple motor (manual, finger) tasks [4,17,55,56]. Recent studies showed that these two phases also occur in practicing stability in a virtual reality environment [8], in practicing maintaining balance on a multiaxial balance board [18] and in a sequential footstep task [31] but task conditions in these studies were quite removed from everyday routines such as walking. The current results show that in the context of practicing the TUG young adults improve significantly in the walking segment of the task; the time-course of accruing these gains across the first session reflects the phases of learning that characterize the acquisition of a new skill.

The gains in TUG duration as well as the changes accrued in practice during the 1st session of TUG practice in some kinematic parameters were subsequently well-retained. We propose that the data support the notion that TUG practice resulted not only in 'online' learning but also in the initiation of 'how to' memory consolidation processes. Evidence for the latter processes occurring 'offline' in the intervals between the TUG sessions is reflected in the robust retention both overnight and over the one-week retention interval, but also in the significant delayed gains (increases) in the mean-step mean acceleration of the trunk in the walking segment of the TUG; acceleration increases that were found to have occurred during the interval between the termination of session-2 and the beginning of session-3.

The notion that during the post-session consolidation phase "slow" qualitative changes occur in the trained task's performance, and the neural machinery serving the execution of the practiced motor task, is suggested by the finding of reductions in performance variability and/ or an improved speed-accuracy relationship across the post-session interval (when no additional practice was afforded) [11,57]. Qualitative changes in the movements' path and in the velocity profiles, indeed the generation of new movement primitives, have also been suggested as the products of consolidation processes [14]. A recent study showed that practicing a sequence of locomotor movements resulted in reduced latencies of step initiation and higher peak velocities during the swing phase [58]. The process of improvement does not necessarily rely only on biomechanical optimization; sequence planning, chunking [14,59,60] and the within-sequence timing of movements, i.e., the temporal structure of task execution, can undergo qualitative changes as well [61–63].

The stabilization of the performance gains attained during a training session, as reflected in an overnight performance assessment and the expression of delayed gains (offline performance enhancement) are considered behavioral outcomes of memory consolidation processes e.g., [24,64,65]. Both outcomes have been described when young adults practiced gross motor performance, in various task conditions [8,18,30,31,66,67]. However, as noted above the participants in these studies practiced tasks and task conditions not commonly encountered in everyday activities. The absence of the delayed gains in TUG performance, after 24 hours, in our study–delayed gains were observed only during the subsequent one-week interval, can be explained by the nature of the TUG, comprised from highly familiar regular movement routines [19]. The delay in the expression of delayed gains in TUG performance may also reflect some retrograde interference to the consolidation of the experience gained in session-1 from everyday activities that came after the TUG practice session [11]. However, the delay can also reflect a continued evolution of consolidation processes after the initial 24 hours post-session [68,69] and the additional practice afforded in session-2. Also, because the gains expected in already familiar tasks may be relatively small [11] the assessment of delayed gains after longer intervals may provide a more sensitive measure [34].

Our analysis also showed that in the interval between sessions 2 and 3 the variability in mean-step mean-acceleration decreased, suggestive of a more stable routine for the walking segment in the context of the TUG by the latter session. A recent review suggests that gait variability should be considered as an indication of a locomotion disorder [70]. However, the current results, showing that variability in gait parameters can learning across repeated gait bouts and that subsequently a minimization of such variability is also possible as a function of practice, suggest the possibility that variability in gait parameters may not necessarily constitute an indication of pathology. The potential for new learning may thus constitute part of the variability in gait parameters seen in typical young adults [1].

Practice related changes in other aspects of the task, specifically in the walking segment of the TUG, were not retained from one session to the next, but this pattern may nevertheless indicate a recalibration between the sessions. Recalibration occurring between the TUG practice sessions should be considered because during the practice session the maximal acceleration of the mean step in the x-axis (lumbar and trunk IMUs) decreased although, concurrently, TUG duration decreased, i.e., task performance became faster with no increase in mean acceleration. This pattern repeated in each of the three sessions; although maximal acceleration of the mean step returned to the level during the initial trials of session-1 TUG duration gains were retained from session to session. Assuming that body mass was unchanged across the sessions, a decrease in acceleration while TUG duration improved, can suggest that walking "efficiency", inversely related to energy expenditure, increased [70].

The current study was not designed to address the question of whether sleep is necessary for consolidating the gains accrued in practice on an every-day task. The conclusion of a recent meta-analysis was that there is a clear but small positive effect on motor skill increase in laboratory tasks if sleep is included in the post learning interval [71]. Whether sleep has a role in the consolidation of 'how to' knowledge in the execution of common, everyday tasks can be directly addressed in future studies; it has been suggested that sleep quality can be a factor in improving walking along an irregular elliptical path [67].

The current results show that in the context of TUG practice, young adults improve significantly in the walking segment of the task during the first session and that these gains are subsequently well-retained. A recent study addressing rehabilitation of mobility in patients with acute or chronic neurological disorders, found that TUG duration improvements often do not reflect improvement in all TUG sub-tasks [45]. Others reported good test-retest reliability for the walking sub-component of the TUG, which suggests little or no learning effect [72]. One possible explanation for the lack of a learning effect in these studies is that the participants (older adults with Parkinson's disease in [72]) were instructed to walk at their normal and comfortable speed whereas in the current study participants were instructed to walk as fast as possible without running. The number of task iterations may be a critical factor, as well; there is evidence suggesting that attaining the plateau phase during a practice session may be necessary for the engagement of consolidation processes, otherwise the gains attained in the session may be likely lost across sessions e,g., [11,16,73].

A recent review [33] emphasized that any experience-dependent improvement in performance can be defined as motor learning. However, the authors argue that while motor learning in many sequence learning paradigms, reflect the learning of how to assemble discrete actions in the correct order, i.e., to anticipate the temporal consistencies in the environment, these paradigms do not advance our understanding of how the execution of the selected actions improve with further training ('motor acuity' tasks [33]). The TUG paradigm can be conceptualized as calling for the assembly pre-existing motor actions into the required motor sequence. However, the current data suggest that what seems to be learned with repeated experience with the TUG task is the way multiple component actions may be each improved to gain 'motor acuity'. Practice-related changes in kinematics can be induced by the additional, special, practice afforded in repeated TUG execution, in complex, sub-tasks for which skill was attained over the life time of the individual.

One should note that participants were made aware of the general aim of study when enrolling in the study, and that the task elements and sequence were explicitly stipulated at the beginning of each session. However, throughout the three sessions participants were given the same instructions—to execute the TUG test as fast as they can without running; participants were not asked to achieve better results than in a previous trial. The participants were not aware that we were looking into whether gains in performance would follow in different phases; nor were they aware of which of the task elements or kinematic measures will be examined. The question of whether knowing the aim of the study was central to modifying the participants' behavior or that similar learning would occur in a more implicit or incidental learning situation is open for further experimentation. Future studies can also address a possible effect of affording feedback to the participants [e.g., 74,75], the interaction with the teacher-instructor [e.g., 76] or the effects of the length of breaks afforded between the trials [e.g., 77].

Overall, the results of the current study indicate that, in young healthy adults, the repeated execution of everyday activities can initiate a new learning process and, moreover, lead to the consolidation of the gains into long-term 'how to' memory. The acquisition of performance gains and their consolidation (retention and refinement) during and following TUG practice

followed the characteristic time-course of learning and retention previously described in the acquisition of new, unfamiliar, laboratory tasks. Practice, it seems, can better old, every-day, tricks even in highly proficient adults.

## Materials and methods

### Participants

The participants were young healthy adults (N = 18; 20–30 years old) with no restriction in walking or sitting due to pain, joint injury or neurologic deficits. Additional exclusion criteria were: surgery or trauma in the past six months; ADHD; active neurologic or psychiatric disorders which require regular medication. Pregnant women were excluded as well.

Participants responded to a demographic and health status questionnaire and a standard questionnaire for determining the preferred foot [78]. The Pittsburgh Sleep Quality Index (PSQI) [79] was used to report the quality, latency, duration, and efficiency of sleep [80]. The Trail Making Test (TMT) [81] was administered to all participants as a measure of cognitive ability; all participants were within the age-appropriate score range norms for Trails A and B.

The participants were recruited via social media; most participants were university students or staff members at the Sheba Medical Center. The participants were paid a small set sum for participation in each session. All participants signed a written informed consent form, as approved by the Ethics Committee of the University of Haifa, and the Human Experimentation Ethics Committee (Helsinki) of the C. Sheba Medical Center. Participants received the original of the consent form, a copy was kept on file at the lab.

Three participants were excluded from analysis due to incomplete compliance in the initial trials of the first session (one participant ran, another stopped short before the turning point, the third failed to keep hands crossed during task execution). Running was defined as locomotion with an instance wherein both feet are off the ground at the same time [82]. Online this was assessed by the experimenter (an experienced observer). Running was formally checked during the analysis of the video-recordings, offline; there was complete correspondence between the observer's online assessment and the video analysis results. Descriptive statistics for the 15 participants included in the analyses are presented in Table 1.

**Table 1. Descriptive statistics of the participants (n = 15).**

| | |
|---|---|
| **Age,** median (range) | 28 (22–29) years |
| **Sex (%)** | 6 (40%); 9 (60%) males; females |
| **BMI[a]**, mean ± SD | 22 ± 2.40 |
| **Years of education,** median (range) | 15 (12–19) |
| **TMT[b] score,** A mean ± SD, B mean ± SD | 25.97 ± 5.24, 52.96 ± 12.61 |
| **PSQI[c] score** (number of participants) | 5, median score 8/15 (53%) participants with a score <5 |
| **Leg preference for object management** (number of participants) | Rt. leg—14/15 (93%); both legs—1/15 (7%) |
| **Leg support preference** (number of participants) | Rt. leg—6/15 (40%); Lt. leg—2/15 (13%); both legs—7/15 (47%) |
| **Hand dominance** (number of participants) | Rt. Hand—14/15 (93%) |

[a]Body Mass Index

[b]Trail Making Test

[c]Pittsburg Sleep Quality Index (a score of 5 and lower is considered good quality sleep).

Group size was determined based on the temporal data from a pilot study, using *i.* video-recordings analysis and *ii.* data derived from the IMUs. The mean and standard deviations of the initial and final four trials of the 1ˢᵗ session, and of the first four trials in session-1 and session-2, were used to assess effect sizes. We calculated the sample size with a) the Simple Interactive Statistical Analysis (SISA) online software (http://www.quantitativeskills.com/sisa/) and b) with the ClinCalc calculator (www.clinicalc.com). According to both calculators, the number of participants needed was 2–22 (median 7–10) depending on the measure used (video, IMUs). Note however that these numbers were derived for comparing across groups; the main comparisons in this study were within-subjects, necessitating smaller numbers of participants.

## Experimental setup and procedure

**The "Timed up and Go" Test (TUG) protocol.**   The TUG test is a clinical test for the assessment of mobility, balance, locomotion and the risk of falling, all relevant to daily function [47]. In the TUG test, the subject starts from a sitting position in a chair with no armrests. The chair is adjusted to achieve a 90˚ flexion of the hips, knees and ankles with the feet flat on the floor and with the subject's back against the backrest. The subject is instructed to get up at a 'go' signal, walk three meters, make a turn, walk back and sit back on the chair. The test provides a time measurement, scored from the 'go' signal until the participant is reseated on the chair. In the current study, participants were instructed to walk as fast as they can, but without running; it was stressed that while speed was essential, running was to be avoided. Before each trial, the participants were asked to lean against the backrest until they hear the cue to start; to keep their hands crossed on the front of their chest throughout the trial; to walk in the instructed path as fast as they can but without running; when returning to the chair and re-seating, to lean against the backrest until they hear the cue that the trial is finished. Participants were asked if they were ready to continue before the initiation of the next trial. Participants were asked not to speak during the trial. The participants did not receive any information about their actual performance during sessions, except for being cautioned not to run but were given general encouragement in the form of "OK/good, when you are ready let's do it once more". (For a representative trial, see S1 Video).

After completing each trial, data were transferred from IMU-level storage and the participants were asked, and indicated, when they were ready for the next trial. The between-trial breaks were on the order of a half a minute ($32 \pm 29$ sec, mean, SD) with one exception—a 19 min break due to technical problem with the IMU.

Six force sensitive sensors were attached to an Arduino type microcontroller and fixed to the surface of the chair on which the participants sat. A voltage threshold of 1.5 Volt was chosen so that the sensors would be sensitive to an averege weight of a seated person. Eight LED lights (LEDs) (4 red, 4 green) were used in order to signal the pressure change. The LEDs color was red when participants were seated and changed to green when participants rose from the seat. The LEDs were placed on the floor by the side of the chair to be captured on the video recordings of the TUG trials.

**Video recording.**   A video camera (I-phone 8, www.apple.com) was used to record the entire sessions and extract the TUG test duration. The video recordings were made at 1080p at 30 frames per second. There was a synchronized audio recording as well, which captured the auditory "go" signal that was used to initiate each TUG trial. The camera was placed on a tripod at a distance of about 1.5 meters behind the participant's chair in order to capture the entire path for the trial's execution and include the LEDs (indicating the pressure on the seat) in the frame.

**Inertial Measurement Unit (IMU).**   A full suite of inertial measurement units (IMUs) (Mobility Lab, APDM, Portland, OR) was used in order to measure TUG duration and

kinematics (Fig 1). The suite included six IMUs on both wrists, both legs, the sternum (trunk) and the lumbar spine (S1 Fig). The APDM IMUs are wristwatch sized wireless devices that capture and store 3-D linear acceleration, angular velocity, and magnetic field (for directional orientation) using onboard accelerometers, gyroscopes, and magnetometers, respectively. Data from the IMUs is derived from three axes; X axis (up—down direction), Y axis (right–left direction) and Z axis (forward-backward direction). The three axes of each IMU are aligned with one another [83]. The data were collected and recorded at the rate of 128Hz. The IMU system was shown to generate accurate and reproducible data for measuring spatiotemporal gait parameters in healthy young adults [83–85]. In this study we report data collected mainly from the trunk and lumbar IMUs.

**Study design.** Participants were instructed to perform the TUG task, repeatedly, in each of the three sessions. The instructions given before each task iteration were identical with the participants required to complete the path (walk) in an anti-clockwise direction (trained direction) (Fig 1A). All session took place in the afternoon hours (2-5pm).

In session-1, participants were asked to perform 18 consecutive iterations of the TUG to test for fast learning and a possible plateau phase. In session-2, 24 hours later, participants were asked to perform eight TUG iterations in the trained direction to test for possible consolidation phase effects. Six or seven days later, in session-3, the participants were asked to perform eight TUG iterations in the trained direction to assess long-term retention. Participants were then instructed to perform a set of eight TUG iterations in a reversed, clockwise, direction, before returning to complete four additional TUG iterations in the trained direction, as the conclusion of session-3. (The results for the reversed-direction TUG trials are not presented in the current paper).

The number of trials (TUG iterations) to be afforded in sessions 1 and 2 was determined in pilot studies. Two possible constraints were taken into account: 1. that a long session may result in fatigue [86]; 2. that a minimal number of task iterations may be necessary for the (potential) induction of skill memory consolidation processes, a point in time reflected in attaining a performance plateau, e.g., [11,16,73]. Pilot study data indicated that performance reached a plateau by trials 8–9 i.e., there were no additional gains in TUG duration in subsequent trials; however there was a small increase the TUG execution duration and the temporal variance (SD) of the TUG duration after the 19[th] trial of session-1, indicating possible fatigue. Thus, session-1 was set to 18 trials. The pilot data also indicated that in session-2, the 4 first trials showed robust retention and little or no fast learning, although there was a small gain between trials 1 and 2 of the session. There were also small additional gains when trials 5–8 were compared to trials 1–4 of session-2.

In cases in which a IMU was dislodged or an unexpected distraction occurred (e.g., someone entering the room where the test took place), the participants were asked to execute an additional TUG iteration as a replacement. Trials in which the participant made the turn earlier than required were not included in the statistical analysis. One participant executed two additional trials (one on session-1 and one on session-2) as replacement trials and another participant executed one replacement trial on session-3.

## Performance measurements

**TUG duration.** The time to complete each TUG iteration was derived, independently, from two sources: I. the video recording of the trial and II. the data from the gyroscope of the participant's trunk IMU. Video based TUG duration assessments and TUG duration according to accelerometers or gyroscopes was shown to be highly correlated [87].

*Video based measurements.* The videos were analyzed using the Wondershare Filmora9 software (version: 9.5.2.10 Wondershare Technology Corp.). The software was used to parse

each second into 30 equal-duration frames; each frame = 0.033 sec. The duration of the TUG trial was calculated from the frame in which the auditory cue, to begin the task, was given, up until the frame in which the participant was reseated in the chair with the back against the backrest (total TUG trial duration).

In addition, the color change of the LED unit, reflecting the pressure sensors in the seat of the chair, as captured in the video recordings, was used to determine the point-in-time (video frame) wherein the participant rose from the seat of the chair, and the point-in-time (video frame) of re-seating, in each trial. The interval was used as an independent measure of TUG trial completion time from rising to re-seating. Because the correlation between the total TUG trial duration and the time from rising to re-seating was robust ($r_s$ = 0.93.9, p<0.01) only the total TUG trial duration is presented in this paper.

A reaction time measure (RT) was defined as the time to initiate the TUG movement sequence after the 'go' signal (video based).

*Gyroscope based measurements.* The TUG start and end cut-off points according to the trunk IMU gyroscope were determined using a Matlab (Matlab vers. R2019b, The Mathworks, Inc., Natick, MA)) based routine (developed by R.G.).

The instantaneous Root Mean Square (RMS) value of the trunk gyro was calculated over a period of 0.1 sec. by:

$$RMS(t_n) = \sqrt{\sum_{i=0}^{N} y^2(t_n - i\Delta t)}$$

Whereas $N = \text{Round}(^{0.1}/_{\Delta}t)$.
The criteria for the two cut-off points, were:

$$t_{start} = \min_t RMS(t_n) > 3.5\sigma_1$$

$$t_{end} = \min_t RMS(t_n) > 3.5\sigma_2$$

$\sigma_1$—the SD of the trunk gyro during the time before start of motion (during the RT period)
$\sigma_2$—the SD of the trunk gyro during the time after end of motion.

The start cutoff (Fig 1C, initial vertical line) was set when the calculated instantaneous RMS value increased to larger than 3.5 times the SD of the trunk gyro during the RT interval. The end cutoff (Fig 1C, rightmost vertical line) was set when the calculated instantaneous RMS value increased to larger than 3.5 times the SD of the trunk gyro during the post trial rest in the seated position, in the reversed time-series.

The resulting time points were individually inspected and, if required, manually corrected. The criterion for manual corrections for the two cut-off points, was a flat line (fluctuations of less than 0.1 radian/sec.) in at least one of the axes for at least 0.05 sec. for both the start and the end time-points. The corresponding data from the gyroscope of the IMUs of both legs, in the Z axis (forward-backward direction) was used as an auxiliary cue for determining cut-off points; the leg gyroscopes often provided long segments (>0.2 sec.) of a flat line (fluctuations of less than 0.1 radian/sec.) at the start and end time-points.

**Mean step kinematics.** Data from the y axis (right–left direction) of the leg gyroscopes was filtered by a low-pass filter (Finite Impulse Response (FIR) order 30, 10Hz bandpass, zero phase filtering). Mid-swing, initial and terminal contact points of each step were identified for both legs based on a previous study [83] (Fig 1D). Step duration for each step, for each leg, was

calculated using the following equation:

$$T_{step}(n) = InitialContactTime(n) - TerminalContact(n)$$

The initial and terminal contact points of each step were used to determine maximal and minimal acceleration from the trunk and lumbar accelerometer data. In addition, the mean acceleration for each step was calculated. Finally, the means for maximal, minimal and mean acceleration in each step were calculated to establish the mean step values in each trial.

## Statistical analysis

Statistical analyses were performed both on TUG duration (video, IMUs) and the kinematic measures derived from the IMUs. All statistical analyses were conducted using IBM SPSS Statistics for Windows, Version 25.0 (SPSS Inc. NY, USA). Each variable was first assessed for normal distribution by the Kolmogorov-Smirnov test. Parametric tests, repeated measures ANOVAs and paired, one-sample or independent t-tests and the Pearson correlation test, were used for measures normally distributed measures. Non parametric tests (Spearman) were used for the very few measures indicated by the Kolmogorov-Smirnov test to have a non-normal distribution.

The relative changes in task initiation time (RT) across pairs of trials in a session were compared using paired t-tests; the overall gains in session-1 were calculated by the difference from the first to the last trial. The offline, between sessions, gains in TUG duration were computed as percent change differences between the final trial of sessions 1 and 2 and the corresponding second trial of the next session, relative to performance in the first trial of session-1. Within-session and between-sessions improvements in terms of TUG duration and kinematics, were assessed by comparing four-trial long segments from the beginning and end of each session. To this end, repeated measures ANOVAs were used with session/segments (time-points) and the trials comprising them as within-subject factors. Within-session gains were assessed by comparing the four-trial long segments from the beginning and end of each session. Between-sessions changes (both overnight and after one week) were assessed by comparing the four-trial long segments from the end of one session and the beginning of the following session. To assess possible differences in the two measures of task execution duration (video, IMUs), a repeated measures ANOVA was used on data from all subjects in all 18 trials of session-1 (trials as a within-subject measure). To evaluate the two phases of performance improvement during session-1, a repeated measures ANOVA was used to compare TUG duration (as a within-subject factor) between the first phase (trials 1–9) and the second phase (trials 10–18) of the session. The Huynh-Feldt adjustments were used, when appropriate, for non-sphericity violation.

The group mean linear and power trend lines were determined using a Matlab algorithm (Matlab vers. R2019b, The Mathworks, Inc., Natick, MA). The difference between the linear trend line slopes and zero was assessed using one sample t-tests.

The correlations between TUG duration measurements according to the video and the IMUs and the correlations of the gains in TUG duration and the change in kinematic measures were assessed by either the Pearson or Spearman's correlation methods depending on whether the data distribution was normal or not, respectively. Descriptive statistics of the participants are presented by a mean (±SD), median (minimum-maximum) or number (percentage).

## Supporting information

**S1 Fig. IMUs for recording kinematics.** A) Placement of the APDM IMUs, the lumbar IMU (attached to the lower back) is shown in fainter color; B) An APDM IMU. (https://apdm.com/

).
(TIF)

**S2 Fig. Performance changes within and between segments/sessions.** A) According to the videos; B) according to the IMUs. */**/***- p<0.05, p<0.01, p<0.001 (repeated measures ANOVA's).
(TIF)

**S1 File. Supplementary results: Practice effects were retained across sessions (video).**
(DOCX)

**S2 File. Duration according to videos data.**
(XLSX)

**S3 File. Duration according to IMUs data.**
(XLSX)

**S4 File. Mean step max. acc. lumbar X data.**
(XLSX)

**S5 File. Mean step max. acc. lumbar Y data.**
(XLSX)

**S6 File. Mean step max. acc. lumbar Z data.**
(XLSX)

**S7 File. Mean step max. acc. trunk X data.**
(XLSX)

**S8 File. Mean step max. acc. trunk Y data.**
(XLSX)

**S9 File. Mean step max. acc. trunk Z data.**
(XLSX)

**S10 File. Mean step mean acc. lumbar X data.**
(XLSX)

**S11 File. Mean step mean acc. trunk X data.**
(XLSX)

**S12 File. Mean step min acc. lumbar X data.**
(XLSX)

**S13 File. Mean step min. acc. trunk X data.**
(XLSX)

**S1 Video. Video recording of a representative trial.**
(MP4)

## Acknowledgments

The authors acknowledge helpful discussion and methodological advice by Tal Krasovsky (PhD), Department of Physical Therapy, University of Haifa.

## Author Contributions

**Conceptualization:** Gil Meir Leizerowitz, Ofer Keren, Avi Karni.

**Data curation:** Gil Meir Leizerowitz, Ran Gabai.

**Formal analysis:** Gil Meir Leizerowitz, Ran Gabai.

**Funding acquisition:** Avi Karni.

**Investigation:** Gil Meir Leizerowitz.

**Methodology:** Gil Meir Leizerowitz, Meir Plotnik, Ofer Keren, Avi Karni.

**Project administration:** Gil Meir Leizerowitz, Avi Karni.

**Software:** Ran Gabai.

**Supervision:** Meir Plotnik, Ofer Keren, Avi Karni.

**Validation:** Ofer Keren, Avi Karni.

**Visualization:** Gil Meir Leizerowitz, Ran Gabai, Avi Karni.

**Writing – original draft:** Gil Meir Leizerowitz, Avi Karni.

**Writing – review & editing:** Gil Meir Leizerowitz, Meir Plotnik, Ofer Keren, Avi Karni.

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
