## [Decision Letter · Decision Letter 0]

27 Sep 2022

PONE-D-22-15151Improving old tricks as new: young adults learn from repeating everyday activitiesPLOS ONE

Dear Dr. Leizerowitz,

Thank you for submitting your manuscript to PLOS ONE. After careful consideration, we feel that it has merit but does not fully meet PLOS ONE’s publication criteria as it currently stands. Therefore, we invite you to submit a revised version of the manuscript that addresses the points raised during the review process.

We look forward to receiving your revised manuscript.

Kind regards,

Peter Andreas Federolf

Academic Editor

PLOS ONE

Journal Requirements:

"The study was supported in part by the Edmond J. Safra, Brain research center, University of Haifa (A.K). The funders had no role in study design, data collection and analysis, decision to publish, or preparation of the manuscript."

"The study was supported in part by the Edmond J. Safra, Brain research center, University of Haifa (A.K). The funders had no role in study design, data collection and analysis, decision to publish, or preparation of the manuscript."

Reviewers' comments:

Reviewer's Responses to Questions

**Comments to the Author**

1. Is the manuscript technically sound, and do the data support the conclusions?

Reviewer #1: Partly

Reviewer #2: Yes

2. Has the statistical analysis been performed appropriately and rigorously? 

Reviewer #1: No

Reviewer #2: Yes

3. Have the authors made all data underlying the findings in their manuscript fully available?

Reviewer #1: Yes

Reviewer #2: Yes

4. Is the manuscript presented in an intelligible fashion and written in standard English?

Reviewer #1: Yes

Reviewer #2: Yes

5. Review Comments to the Author

Reviewer #1: The present study aims at investigating potential motor learning when practicing a familiar movement in young healthy adults. For this purpose, the authors used the Timed-Up-and-Go (TUG) task and analyzed learning gains using video recordings as well as IMU data. The authors conclude that common daily activities, such as the TUG task, can be learned with a similar pattern compared to novel motor skill acquisition.

In general, the study addresses a relevant topic in the field of motor learning since most experiments focus on de novo learning in relatively simple fine-motor tasks. The manuscript is well written and (in most parts) clearly structured and presented. On the other hand, I have several major concerns regarding the motor task, the way the task was administered and analyzed, and the reporting of the statistical analysis (see specific comments).

Specific comments:

Introduction:

- Overall this part is well written and comprehensible. The authors clearly describe the time course of motor learning and the need for investigating more complex ecologically valid motor tasks. However, I would appreciate if the authors could elaborate on why learning curves of everyday motor tasks should be different from typical laboratory motor tasks. Are there any neurophysiological data or hypotheses may explaining this?

Methods:

- Did the authors preregister their study including the hypotheses, study design and analyses?

- l. 413-418: How did the authors ensure that including 15 participants is sufficient to answer their question? Was the sample size based on an a-priori power calculation or previous experiments? How were the participants recruited and were they blinded to the authors` hypotheses or aim of the study?

- Even though I understand the authors’ rational for using the TUG task I’m somewhat hesitant whether practicing the TUG really represents a motor tasks that differs from other more complex gross-motor tasks already used in previous experiments, such as balancing or walking. Learning gains are highly task-specific. Even though standing up and walking is a common everyday activity in young adults (similar to balancing or typing on a keyboard) the specific execution of the task (i.e. motor sequence) might be new for the participants (e.g. standing up and walking a specific distance and turning as fast as possible), and thus might represent de novo learning. I would like if the authors could explain why practicing the TUG should be different from other everyday tasks studied in laboratory conditions (e.g. balancing, typing on a keyboard, etc.) as mentioned by the authors?

- l. 450: How was running defined and how did the authors check for running?

- l. 481-484: Did the authors use standardized instructions? It could be helpful for pervious experiments to provide them in the supplementary material. Since knowledge of result has shown to be important for the acquisition of motor skills did the participants receive any information about their results during practice. Did the authors provide knowledge of performance or provide any additional information?

- l.486-491: Why did the authors decide to perform 18 trials during memory encoding and 8 trials during retention tests? I think, the number of trials could significantly impact the results of the study, and thus should be clearly justified. Why are 18 trials enough for memory encoding in this type of task? Why did the authors perform 8 trials during retention test? I assume that performing 8 trials during one-day retention test may induce additional practice (i.e. memory reconsolidation or reactivation of the motor engram) potentially affecting the meaningfulness of long-term retention test after 7 days. It seems that the number of trials are arbitrary which in my opinion is a major concern and limits the value of the experiment. I would appreciate if the authors could elaborate on this.

- I’m missing any information on the duration of breaks between trials during encoding and retention tests. This again could dramatically affect the results particularly if not standardized. Furthermore, missing information on that makes it impossible to replicate the study.

- Did the authors check for other factors potentially affecting motor learning in addition to sleep (e.g. caffeine intake, physical activity, etc.)?

- l. 494: Why are the results of the transfer test not reported?

- l.494-499: How many participants executed one or more additional trials? I think this needs to be reported. If performing only 18 practice trials even one additional trial could influence the learning curve, and thus makes it difficult to compare the data.

- l. 533-538: What was the rational for using a SD larger than 3.5 as cutoff. Is this cutoff based on previous experiments?

- General comment statistical analysis: In general, the description of the statistical analysis is very vague. Many analysis reported in the results section are not mentioned in the statistical analysis section (see comments “results”). Please specify the exact method and data you used for analysis (e.g. mixed ANOVA, between- and within-group variables, etc.). Otherwise it is impossible to replicate the experiment. Further, did the authors check for normality, variance homogeneity and sphericity of the data where appropriate? What method was used for detecting outliers and how were they treated?

- l. 563: Why did the authors use Spearman correlation instead of Pearson correlation to compare TUG duration between video recording and IMU data? If data was not normally distributed why did the authors use ANOVA to analyze data?

- l. 563-570: The authors used only one trial (i.e. first and last trial) to calculate relative improvements in session 1 (i.e. within-session improvement). In contrast, in the next sentence the authors state that they used 4 trials to analyze within- and between session changes. Finally, session 1 was divided in two phases each consisting of 8 trials. Again the number of trials used for analysis seems to be arbitrary. Please specify why you used this specific number of trials for each analysis (e.g. is 1 trial or are 4 trials appropriate to account for motor fluctuations while not masking learning gains).

- l. 570-571: I would appreciate if the authors could elaborate on why it is meaningful to correlate offline learning changes with PSQI values in this context. The PSQI retrospectively includes average sleep behavior of the last four weeks. This may not reflect sleep quality of the night between encoding and retention sessions.

Results:

- In general, the results are clearly structured. However, most of the statistical analyses are not mentioned in the methods sections. This makes it difficult to follow and almost impossible to understand the way how data was analyzed. In my opinion, this is a major issue of the manuscript. For example, in l. 159-162 the authors report the performance improvements in session 1. Statistics indicate that the authors used a repeated-measures ANOVA for analysis including only the first and last trial (e.g. one degree of freedom). It is not clear why they used ANOVA including only two variables instead of paired sample t-test or alternatively why they didn’t incorporate all practice trials of session 1 in the ANOVA. Further examples are: l. 164-166 (in the methods section the authors only mention that video and sensor data are compared using Spearman correlation), l. 167-172 (information on the method and analysis of reaction time is completely missing), l. 173-175 & 182-184 (no information on the slope and power function analysis is provided in the methods section), there is no information on the statistical analysis of step kinematics in the methods section, etc.

- As aforementioned in methods section, I’m missing information on assumption checks for the statistical tests.

- Closely related to this comment, did the authors check for normal distribution of the TUG data or adjust for baseline performance? I would assume that performance in such a task could be highly variable between participants due to previous experience.

- l. 184-190: I would like if the authors could elaborate on how and why they decided to split the learning curve of session 1 in two phases (i.e. trial 1-9 & 10-18)? Was their any mathematical or statistical rationale? Did the authors check for best model fit (e.g. using different number of trials)?

- l. 190-191: Please rephrase.

- l. 207-208: Please provide statistics.

- I think, it would be helpful for the readers if the authors provide the data of the offline changes (e.g. in percent like they did for online gains).

Discussion:

- The discussion is clearly structed and good to follow. However, I would rather like if the authors could primarily elaborate on why the TUG task, the way the task was administered (e.g. number of practice trials, break duration, feedback, etc.), and the analysis (e.g. splitting session 1 in two phases, number of trials used for analysis, etc.) is appropriate to answer the research question and to draw these conclusions (see my previous comments).

Figures:

- I think, including bar charts showing the within- and between session changes (i.e. online and offline changes) would improve the quality of the figures.

Reviewer #2: In this study, the authors investigated the short-term and long-term effects of repeated executions of the Timed-Up-and-Go (TUG) task (a clinical test of daily function), a task comprised from a sequence of motor actions: rising from a seated position, walking, turning, and returning to a seated position. The study participants (N=15) were young healthy adults who practiced the TUG task on the 1st day 18 times and retested 24 hours and 1 week post-practice. Performance characteristics were derived from video recordings and 5 wearables with gyroscope and accelerometers. The results showed that the duration of TUG execution was shortened, and that the trial-to-trial changes were fitted by a power function. Further off-line gains in performance were obtained in the subsequent test-sessions. Different kinematic parameters followed different time-course of practice-related changes: some were related to within-session, while other to between-session changes. The authors conclude that their results show that even common everyday motor tasks can be improved by practice.

This is an interesting manuscript with many strengths, it supplies novel, clinically relevant, and theoretically important results. Specifically, the study adds new knowledge to the theories of skill acquisition and maintenance, generally assumed to be relevant mainly to the learning of new motor skills and/or task conditions. The manuscript presents sound methodology and analysis, however the theoretical basis (the structure of the literature review) and the rational/hypotheses are not clear enough. Therefore, the manuscript needs several clarifications and revisions before I can endorse the publication. My points are listed below.

Technical points:

1. Please, incorporate figures and legends in the text. The old-style arrangement of the figures at the end of the document is very inconvenient for reading from PDF (as opposed to printed manuscript).

2. The resolution of the Figures is very low, please, improve their quality. In the current version the axis names are barely seen.

3. The style and the size of the font are inconsistent. Please, check.

Major points:

4. Line 65: I would expect here a sentence about deliberate practice models, as they refer to long-term improvements in non-novel tasks.

5. Lines 83-95: The general message of the paragraph is unclear. Is it about laboratory vs. ecological settings, about generalization of principles of learning simple tasks vs. complex tasks or about the determinants of long-term motor task presentation? Balance tasks are not necessarily more complex than sequence task. I suggest avoiding “complexity” issues or providing a clear operational definition, if complexity is a key construct the authors are interested to discuss. In my view, a simple taxonomy, sequence vs. adaptation motor tasks is enough.

6. Lines 92-95: I find it necessary to include the type of instruction and/or feedback as important determinants of the course of sequence learning (e.g., self-paced, cue-triggered performance sequence-of-movements / with or without knowledge of results).

7. Lines 103-110: Wouldn’t it be more precise to use the term “motor actions” instead of “activities”?

8. Lines 107-126: What is surprising here? Classical paradigms of sequence learning are based on practicing a specific order of finger/arm movements. The basic elements, e.g., finger-to-keyboard movement, are well-trained by definition. The TUG task also focuses on a specific sequence of well-trained (this time, whole-body) motor actions with novel task demands (perform as fast as possible, but don’t run). As well, the TUG task demands to keep the hands on the chest while walking – an uncommon motor constraint that can’t be considered a “highly familiar sub-task and routines”. Please, clarify this point in the introduction and in the discussion.

9. Lines 127-132: Please, include specific hypotheses of the study.

10. In the Methods, please, include justification for the sample size.

11. In the Methods, what does it mean: “The participants were partly reimbursed for taking part in the study.”?

12. In the Methods, please, clearly state what type of feedback was provided to the participants.

13. It would be helpful to have a video recording of a representative trial presented in the supplementary information.

14. Lines 173-175: “with the linear trend line slopes of the participants” – please, rephrase. The slopes are individually fitted to the RTs across trials for each participant, I guess.

15. Line 222: Why the authors substitute “TUG duration” by “speed”? Please, justify or use a consistent name for the dependent variable.

16. Lines 231-236: As transfer tests are excluded from the current paper, but could affect the performance of the trained TUG direction, I suggest excluding the report and discussion of additional gains in TUG duration accrued across session-3. Please, present the per cent of improvement from the baseline to the initial, pre-transfer test trials of session-3.

17. Lines 317-326: Please discuss considering the existing literature in sequence learning referring to the distinct within-session and between-session changes in kinematic characteristics of individual movements, - where different kinematic characteristics show a distinct time course and a specific susceptibility to interference.

6. PLOS authors have the option to publish the peer review history of their article (what does this mean?). If published, this will include your full peer review and any attached files.

Reviewer #1: No

Reviewer #2: No

---

## [Author Response · Author response to Decision Letter 0]

17 Jan 2023

January 17th. 2023

To: Dr. Peter Andreas Federolf   

Ref: PONE-D-22-15151

Title: Improving old tricks as new: young adults learn from repeating everyday activities

Dear Dr. Federolf, 

Enclosed please find our revised manuscript: "Improving old tricks as new: young adults learn from repeating everyday activities", an original research paper, reviewed as PONE-D-22-15151.

We thank you and the reviewers for the thoughtful reading of the manuscript and for the positive and helpful comments. We are very happy to note that the reviewers found our manuscript addresses 'a relevant topic in the field of motor learning'. We have carefully considered the comments and suggestions made by the two reviewers and we have used these comments to revise the paper; we are confident that the revised version now presents our findings in a clearer manner and hope that you will find the paper suitable for publication in PLOS ONE. 

We thank you for the opportunity to revise the manuscript. In the following we explain how we addressed the major points, summarized in your notes and comments and the comments and suggestions made by the reviewers, as point-by-point responses.

1R: We ensured our manuscript meets PLOS ONE's body formatting guidelines. We changed the heading, figures and tables citations as recommended in the style requirements. Affiliations are now provided according to the affiliations formatting guidelines. 

2R: As requested, in your letter of the 27th Sept, we now provide additional details regarding participant consent in the ethics statement in the Methods and in the ‘online submission information’. 

In the statement concerning participants’ consent we now write: "All participants signed a written informed consent form, as approved by the Ethics Committee of the University of Haifa, and the Human Experimentation Ethics Committee (Helsinki) of the C. Sheba Medical Center. Participants received the original of the consent form, a copy was kept on file at the lab.”

3R. The reference to the funding of the study was deleted from the acknowledgements section of our manuscript. We ask you to change the online submission form on our behalf. In addition, we amended the statement in our new cover letter as requested.

Information about funding is now stated only in the Funding Statement. We added the grants number: "The study was supported in part by the Edmond J. Safra, Brain research center, University of Haifa (grant 100007648/18 to A.K). The funders had no role in the design of the study, data collection and analysis, decision to publish, or preparation of the manuscript."

4R. Data Availability: The anonymized data base have been made public in Open Science Framework (https://osf.io/) - DOI 10.17605/OSF.IO/XVSQ4. In addition, our data were uploaded as a Supporting Information files. 

Please update out Data Availability statement on our behalf to reflect the information we provide. 

5R. The E. J. Safra foundation’s grant number 100007648/18 was added to the Funding Statement.

Sincerely,

Gil Leizerowitz, Ran Gabai, Meir Plotnik, Ofer Keren, Avi Karni

 

The following are point-by-point responses to the reviewers’ comments and suggestions: 

Reviewer #1:

 The present study aims at investigating potential motor learning when practicing a familiar movement in young healthy adults. For this purpose, the authors used the Timed-Up-and-Go (TUG) task and analyzed learning gains using video recordings as well as IMU data. The authors conclude that common daily activities, such as the TUG task, can be learned with a similar pattern compared to novel motor skill acquisition.

In general, the study addresses a relevant topic in the field of motor learning since most experiments focus on de novo learning in relatively simple fine-motor tasks. The manuscript is well written and (in most parts) clearly structured and presented. On the other hand, I have several major concerns regarding the motor task, the way the task was administered and analyzed, and the reporting of the statistical analysis (see specific comments).

 Response: We thank the reviewer for the helpful comments and suggestions. We appreciate the reviewer’s assessment that “The manuscript is well written and (in most parts) clearly structured and presented.” We have carefully considered the comments and suggestions made by the reviewer and are confident that the revised manuscript now addresses these points and by incorporating in the revised manuscript additional information and clarifications, as suggested by the reviwer, the clarity of our paper is improved. In the following we explain how we addressed the major points raised in your specific comments and suggestions in a point-by-point manner.

Specific comments:

Introduction:

Comment - Overall this part is well written and comprehensible. The authors clearly describe the time course of motor learning and the need for investigating more complex ecologically valid motor tasks. However, I would appreciate if the authors could elaborate on why learning curves of everyday motor tasks should be different from typical laboratory motor tasks. Are there any neurophysiological data or hypotheses may explaining this?

R: We appreciate the reviewer’s comment about the writing and clarity. The reviewer raises the question of why learning curves of everyday motor tasks should be different from typical laboratory motor tasks. We do not think that there is a reason for assuming that learning in real life situations and in the lab would differ in a fundamental way (with some caveats concerning the relative complexity of tasks); however, there is a very different pattern of performance changes across task iterations (practice effects) in novel tasks (“de novo” learning) and in well-rehearsed tasks. 

The difference is clear from studies of laboratory tasks across multiple sessions: the learning curves one finds in the early sessions, specifically the 1st learning-practice session (in both perceptual and motor tasks), markedly differs from the learning curves one finds in later sessions. Indeed, it has been suggested that the characteristics of the learning curves across task repetitions can be used as indicators of whether a task is novel or a previously well-trained one (novelty effect; e.g Korman, Raz, Flash, & Karni, 2003).

In the real world, we do not expect that being requested to walk quickly in a corridor in a lab (with no obstacles) would show gains in performance characterizing the learning of a new task or cause significant changes in the person’s subsequent walking routines, because a highly trained routine, as walking, is expected to show ‘transfer’ and, if at all, some small adaptation gains between the first and the second or perhaps even the second and third trials rather than to show a “fast learning” phase as in the performance of a novel task. We expect ‘expert’ walkers to walk in a corridor using a set of well-established routines. 

Laboratory tasks, both perceptual and motor, show consistently that the within-session improvements in performance that occur across repeated task iterations minimize or disappear beyond the second or third training sessions. In fact, in laboratory setting, there are no significant within-session gains after the tasks has been trained for a few sessions (except small readaptation gains between first and second iterations in the later sessions) (Gal, Gabitov, Maaravi-Hesseg, Karni, & Korman, 2019; Karni & Sagi, 1993; Korman et al., 2007, 2003).

 So the questions addressed are whether we should expect to see 'fast learning' occurring in well-rehearsed (familiar) tasks (rather than fast adaptation (Krakauer, Hadjiosif, Xu, Wong, & Haith, 2019; Shadmehr, Smith, & Krakauer, 2010; Yang, Cowan, & Haith, 2021), or effective transfer (Rosalie & Müller, 2014)? Second, can we see evidence for consolidation phase (post-session) ‘delayed’ gains (Korman et al., 2003; Telgen, Parvin, & Diedrichsen, 2014)? For example, would one expect a professional basketball team to undergo a new learning process (relating to ball shooting) each time they switch to a new court? In our current study, we show not only performance improvement (task execution times) but changes (that persist and build up beyond the session, i.e., a pattern of delayed gains) in the kinematics of the walking segment of the TUG, in young healthy participants (for theoretical perspectives see (Krakauer et al., 2019; Yang et al., 2021). 

 It has been suggested that different neuronal mechanisms underlie the recalibration of an existing versus acquisition of a new task performance controller mechanism and the generation of a new performance routine (‘de novo’ learning) (Karni, 1996; Krakauer et al., 2019). A fast learning phase in the initial learning session (Karni & Sagi, 1993; Korman et al, 2003), a between-sessions shift in speed-accuracy tradeoff (Korman et al., 2003; Telgen et al., 2014) and subsequently long-term, “offline”, gains between sessions (Karni & Sagi, 1993; Korman et al., 2003; Telgen et al., 2014) have been suggested as characteristic of the ‘de novo’ building of a task specific controller and the generation of a new performance routine. 

 Given the reviewer’s comment we have extended the following explanation of the issue in the introduction: “Because the characteristic phases in the mastering of new tasks can be delineated in, for example, perceptual skills as well (Karni & Sagi, 1993) it has been proposed that the acquisition of a new skill (i.e., the generation of long-lasting practice-dependent gains in ability) reflects a common repertoire of basic neural processes, albeit in the different brain systems that subserve the execution of the different tasks (e.g.,Karni, 1996; Karni et al., 1998). Thus, the time-course of skill acquisition rather than the specifics of the task may best reflect processes of neuronal and systems level brain plasticity(Dudai, Karni, & Born, 2015; Korman et al., 2003; Telgen et al., 2014). This would suggest that if young adults do acquire new or additional skill when afforded the opportunity to practice common everyday activities, the time-course of performance improvement would follow that of new skill learning in laboratory tasks and conditions. However, we do not expect that the characteristic phases of new learning will replicate in a task which is already over-trained, as in walking or typing. In other words, at issue is the question of whether we should expect to see the characteristic time-course of new skill acquisition (rather than fast adaptation (Krakauer et al., 2019; Shadmehr et al., 2010; Telgen et al., 2014), and effective transfer (Rosalie & Müller, 2014) occurring in well-rehearsed (familiar) tasks such as, for example, typing when we introduce a new text or raising from an unfamiliar chair. Viewed from a related framework there is the notion that very extensive, long-term and deliberate practice is required to gain, albeit via incremental gains, in order to attain expertise in well rehearsed tasks in different domains (Ericsson, Krampe, & Tesch-Römer, 1993). Thus the question would be whether, if afforded an opportunity for deliberate practice, young adults would show a pattern of continued incremental improvement or whether a deliberate-structured practice opportunity would result in initiating a form of ‘de novo’ learning albeit for a previously highly practiced, everyday task (Baker & Young, 2014; Ericsson & Harwell, 2019; Krakauer et al., 2019; Yang et al., 2021)."

 It has been suggested that different neuronal mechanisms underlie the recalibration of an existing skill or sets of skills, versus the acquisition of a new task performance controller mechanism and the generation of a new performance routine (‘de novo’ learning) for the trained task (Karni, 1996; Krakauer et al., 2019). A fast learning phase in the initial learning session (Karni & Sagi, 1993; Korman et al., 2003), a between-sessions shift in speed-accuracy tradeoff (Korman et al., 2003; Telgen et al., 2014) and subsequently long-term, “offline”, gains between sessions (Karni & Sagi, 1993; Korman et al., 2003; Telgen et al., 2014) have been suggested as characteristic of the ‘de novo’ building of a task specific controller and the generation of a new performance routine.” 

Methods:

C - Did the authors preregister their study including the hypotheses, study design and analyses? 

R: An early plan (hypotheses, study design) of the study was pre-registered in the OSF (Open Science Framework (OSF) - https://osf.io/) as part of the process of submitting the study to approval by the ethics committee. Two study plans were uploaded to the site (the preliminary plan from the year 2019 and the current article draft. The study is not made open to the general public. The data relevant to the currents study are uploaded in OSF according to the PLOS ONE publication policy.

C - l. 413-418: How did the authors ensure that including 15 participants is sufficient to answer their question? Was the sample size based on an a-priori power calculation or previous experiments? How were the participants recruited and were they blinded to the authors` hypotheses or aim of the study? 

R: Prior to the experiment we conducted a pilot study on two individuals (not included in the participants reported in the paper). These participants underwent an identical training protocol in the first session: 20 iterations of the Timed Up and Go (TUG) test. In the second session, 24 hrs. later, Subject A was asked to perform 21 additional trials, whereas Subject B was asked to perform eight TUG trials in the previously trained direction (anti-clockwise), then eight clockwise TUG trials, and then four additional anti-clockwise TUG trials (the protocol used in this study). The sample size was determined after calculating and comparing the mean ± SD of the first four trials and the last four trials in the first session and by comparing the first four trials in the first session and the first four trials in the second, overnight, session. The temporal data used was 1. video-recordings based and 2. derived from the IMUs. We calculated the sample size with a) the Simple Interactive Statistical Analysis (SISA) online software (http://www.quantitativeskills.com/sisa/) and b) with the calculator provided in the web site of the ClinCalc company (www.clinicalc.com). According to both clinical calculators, the number of participants needed was 2 – 22 (median 7-10) depending on the measure used (video, IMUs). Note however that these numbers were derived for comparing across groups; the main comparisons in this study are within subjects, and thus smaller numbers should suffice. 

 In addition, in the scientific background section we reviewed studies that examined the time course and constraints of acquisition and consolidation of motor and perceptual tasks; in most publications the number of subjects included in each group was 5 - 15 participants (Brown, Robertson, & Press, 2009; Cirillo, Todd, & Semmler, 2011; Karni et al., 1995, 1998; Karni & Sagi, 1991; Meital, Korinth, & Karni, 2013; Press, Casement, Pascual-Leone, & Robertson, 2005; Walker, Brakefield, Morgan, Hobson, & Stickgold, 2002). Thus, a sample size of 15 participants was deemed reasonable.

The participants were recruited via social media; the majority were students or staff members at the Sheba Medical Center. 

The participants were aware of the general aim of the study – to test whether new learning occurs in everyday tasks. 

This was added to the Participants section: 

“Group size was determined based on the temporal data from a pilot study, using i. video-recordings analysis and ii. data derived from the IMUs. The mean and standard deviations of the initial and final four trials of the 1st session, and of the first four trials in session-1 and session-2, were used to assess effect sizes. We calculated the sample size with a) the Simple Interactive Statistical Analysis (SISA) online software (http://www.quantitativeskills.com/sisa/) and b) with the ClinCalc calculator (www.clinicalc.com). According to both calculators, the number of participants needed was 2 – 22 (median 7-10) depending on the measure used (video, IMUs). Note however that these numbers were derived for comparing across groups; the main comparisons in this study were within-subjects, necessitating smaller numbers of participants.”

C - Even though I understand the authors’ rational for using the TUG task I’m somewhat hesitant whether practicing the TUG really represents a motor task that differs from other more complex gross-motor tasks already used in previous experiments, such as balancing or walking. Learning gains are highly task-specific. Even though standing up and walking is a common everyday activity in young adults (similar to balancing or typing on a keyboard) the specific execution of the task (i.e. motor sequence) might be new for the participants (e.g. standing up and walking a specific distance and turning as fast as possible), and thus might represent de novo learning. I would like if the authors could explain why practicing the TUG should be different from other everyday tasks studied in laboratory conditions (e.g., balancing, typing on a keyboard, etc.) as mentioned by the authors?

R: We understand the concern about the difference between the TUG and other complex gross motor tasks that were tested in previous experiments. The point is the ‘distance’ between every-day and laboratory test conditions. New learning, rather than transfer, can potentially occur if ‘the task’ is applied to novel non-standard conditions in the laboratory. This is what has been the case for example in Elion et al, 2015. 

In the clinical literature there is a strong assumption that TUG performance is stable in healthy adults, young or old, because it well-represents (replicates) prevalent everyday experience (everyday conditions). In fact, there is a set of age adjusted standards for TUG performance. And indeed, good test - retest reproducibility (reliability of repeated performance measures) was shown in, for example community dwelling elderly, with very high intra-class correlations (icc = 0.97, 0.99) (Podsiadlo & Richardson, 1991; Steffen, Hacker, & Mollinger, 2002). So, given these results, one would not expect new skill learning process in repeating (training) the TUG. 

The reviewer suggests two examples, balance training and typing on a keyboard. The testing of balance skill acquisition by healthy participants in laboratory settings do not relate to standard everyday activities; the studies often address extreme and unfamiliar balance challenges in unfamiliar contexts (e.g., Elion et al., 2015) alike to challenging surfing conditions; (Valle, Casabona, Cavallaro, Castorina, & Cioni, 2015). The postural control which is required during walking and turning is different from that required in standing (Segal, Orendurff, Czerniecki, Shofer, & Klute, 2008; Winter D.A, 1995).

None of the typing tasks, addressed in research on sequence learning, employ the standard hand-finger positions of standard (blind) typing (again because of the assumption that this typing mode has been well or “over” learned) – so not related to everyday experience. To our knowledge, there is no study which examines learning in typing task conditions in which participants are already over-trained (except as control conditions). Still, the issue is as the reviewer suggests, whether and through what mechanisms is a new skill established, given that (in fact) in every laboratory paradigm, the task to be learned and mastered is often composed from pre-existing sub-skills. The question, thus, is what improves in practicing the "new" task demands and conditions (or what is new for the motor system, rather than what is considered new by the trainee). The most widely used paradigm (task) for studying motor skill learning and especially motor sequence learning is the Serial Reaction Time (SRT) - a finger tapping skill, a series of key presses, according to a given set of visual cues. It is clear that multiple sub-skills are involved: e.g., key presses by individual fingers, hand-eye coordination, spatial mapping. This issue however is not addressed in SRT studies. In fact, in most studies this is not discussed. The current study provides data toward resolving the question of what happens in practice on a new ‘problem’ or settings to sub-skills that are already performed with very stable (ceiling, "over-learned") level of performance.

The SRT is predicated on an assumption that little learning if at all occurs in the random condition, when no sequence is present in the input-output (Nissen & Bullemer, 1987). Indeed, when typical participants (even children) are given the random condition the learning involved in repeatedly executing the task is minimal and much smaller than learning induced by tapping repeated (implicit) sequences (Desmottes, Meulemans, & Maillart, 2016). 

Typing sequences is not considered a good model for complex gross-motor tasks; a recent study suggests that Sequential Finger Tapping Tasks do not well-represent the sequential whole-body movements typically encountered in everyday life (Christova, Aftenberger, Nardone, & Gallasch, 2018; Freitas, Saimpont, Blache, & Debarnot, 2020). Gross movements involve larger body segments and require more complex muscle synergies including postural stabilization and anticipatory adjustment (Chiou, Hurry, Reed, Quek, & Strutton, 2018; Chiou et al., 2015). Furthermore, training of everyday motor task involves large muscle groups, which may lead to muscle fatigue and physical exhaustion before such effects are expected for finger movements (Zając, Chalimoniuk, Gołas̈, Lngfort, & Maszczyk, 2015). 

C - l. 450: How was running defined and how did the authors check for running?

R: Running was defined as locomotion with an instance wherein both feet are off the ground at the same time (Novacheck, 1998). Online this was assessed by the experimenter (an experienced observer; clinician using the TUG). Running was formally checked during the analysis of the video-recordings. If running occurred the trial was discarded from further analysis (there was a complete correspondence between the observer's online assessment and the video analysis results). One participant was not included in the analysis because of running.

This is now clarified in the Methods section: “Three participants were excluded from analysis due to incomplete compliance in the initial trials of the first session (one participant ran, another stopped short before the turning point, the third failed to keep hands crossed during task execution). Running was defined as locomotion with an instance wherein both feet are off the ground at the same time (Novacheck, 1998). Online this was assessed by the experimenter (an experienced observer). Running was formally checked during the analysis of the video-recordings, offline; there was complete correspondence between the observer's online assessment and the video analysis results. Descriptive statistics for the 15 participants included in the analyses are presented in Table 1."

C - l. 481-484: Did the authors use standardized instructions? It could be helpful for pervious experiments to provide them in the supplementary material. Since knowledge of result has shown to be important for the acquisition of motor skills did the participants receive any information about their results during practice. Did the authors provide knowledge of performance or provide any additional information?

R: We used standardized instructions. The participants were instructed to lean against the backrest until they hear the cue to start; to walk in the instructed path as fast as they can but without running; to keep their hands crossed on the front of their chest throughout the trial; to lean against the backrest when returning to the chair until they hear the cue that the trial is finished. Participants were asked if they were ready to continue before the initiation of the next trial. Participants were asked not to speak during the trial. 

Participants did not receive any information about their actual performance during sessions, except for being cautioned not to run, but were given general encouragement in the form of “OK/good, when you are ready let’s do it once more". 

This is now clarified in the Methods section: “In the current study, participants were instructed to walk as fast as they can, but without running; it was stressed that while speed was essential, running was to be avoided. Before each trial, the participants were asked to lean against the backrest until they hear the cue to start; to keep their hands crossed on the front of their chest throughout the trial; to walk in the instructed path as fast as they can but without running; when returning to the chair and re-seating, to lean against the backrest until they hear the cue that the trial is finished. They were asked if they were ready to continue before the initiation of the next trial. Participants were asked not to speak during the trial. The participants did not receive any information about their actual performance during sessions, except for being cautioned not to run, but were given general encouragement in the form of “OK/good, when you are ready let’s do it once more”. (For a representative trial, see Supplementary Material - video)."

C - l.486-491: Why did the authors decide to perform 18 trials during memory encoding and 8 trials during retention tests? I think, the number of trials could significantly impact the results of the study, and thus should be clearly justified. Why are 18 trials enough for memory encoding in this type of task? Why did the authors perform 8 trials during retention test? I assume that performing 8 trials during one-day retention test may induce additional practice (i.e. memory reconsolidation or reactivation of the motor engram) potentially affecting the meaningfulness of long-term retention test after 7 days. It seems that the number of trials are arbitrary which in my opinion is a major concern and limits the value of the experiment. I would appreciate if the authors could elaborate on this.

R: All these points (pertaining to session length, number of session trials) were explicitly addressed in pilot studies. We considered two possible constraints: 1. that a longer session may result in fatigue; 2. that a minimal number of task iterations is a necessary condition for the (potential) induction of skill memory consolidation processes (e.g., Korman et al, 2003; Hauptmann and Karni, 2002; Hauptmann et al, 2005). In the pilot study we used 20 iterations in the first session. We found that the TUG execution duration and the temporal variance (SD) of the TUG duration in the younger participant increased in the last two trials. On the other hand it was clear from the pilot data (young and older participants) that by trial 9 and onwards there were no additional gains in TUG duration. Thus, we decided to shorten session-1 to 18 trials. In the pilot study we also used 20 trials in session-2. We noted that the 4 first trials showed very robust retention and little or no fast learning; there was a small gain between trial 1 and 2. However, we also found that eight consecutive trials of session-2 induced additional practice; this was shown by a small additional improvement in the trials 5-8 compared to trials 1-4 of the session. 

All instances of testing by necessity provide additional opportunity for practice. However, it is not true that once beyond the fast-learning phase and into the plateau phase participants long-term gains are sensitive to the number of trials afforded at the later part of the session; the concern is for fatigue effects accruing (online). 

Given the reviewer’s comment we now clarify that the number of trials in the sessions was the product of a pilot study and careful consideration of the double constraints of fatigue and the need for satisfying a minimal task iterations. We write:

"The number of trials (TUG iterations) to be afforded in sessions 1 and 2 was determined in pilot studies. Two possible constraints were taken into account: 1. that a long session may result in fatigue (Zając et al., 2015); 2. that a minimal number of task iterations may be necessary for the (potential) induction of skill memory consolidation processes, a point in time reflected in attaining a performance plateau, (e.g., Hauptmann & Karni, 2002; Hauptmann, Reinhart, Brandt, & Karni, 2005; Korman et al., 2003). Pilot study data indicated that performance reached a plateau by trials 8-9 i.e., there were no additional gains in TUG duration in subsequent trials; however there was a small increase the TUG execution duration and the temporal variance (SD) of the TUG duration after the 19th trial of session-1, indicating possible fatigue. Thus, session-1 was set to 18 trials. The pilot data also indicated that in session-2, the 4 first trials showed robust retention and little or no fast learning, although there was a small gain between trials 1 and 2 of the session. There were also small additional gains when trials 5-8 were compared to trials 1-4 of session-2.”

C - I’m missing any information on the duration of breaks between trials during encoding and retention tests. This again could dramatically affect the results particularly if not standardized. Furthermore, missing information on that makes it impossible to replicate the study. 

R: After each trial there was a brief break in which data transfer (IMUs to PC) was done and the subject was asked if s/he needed time or was ready to continue. The median time between the beginning of one trial and the beginning of the next one was 23 sec. (15 sec. – 19 min, an instance of a IMU problem). The mean break duration was 32 ± 53 sec. (SD). 

Given the reviewer’s comment we now added to the text: “After completing each trial data were transferred from IMU-level storage and the participants were asked, and indicated, when they were ready for the next trial. The between-trial breaks were on the order of a half a minute (32 ± 53 sec, mean, SD).”

C - Did the authors check for other factors potentially affecting motor learning in addition to sleep (e.g., caffeine intake, physical activity, etc.)?

R: We asked participants, in a demographics and health condition questionnaire, about their consumption of beverages containing caffeine (1.24±1.03 cups per day). Participants were asked to keep their caffeine intake during the experiment at their usual levels. 12 of the 15 participants reported they were undertaking physical/sports activity on a regular basis (during 2-25 recent years (median – 10 years). 

At the beginning of session-2 participants were asked about physical activity during the 24 hours between-sessions interval. 

No significant correlations were found between the participants’ daily caffeine consumption (1.24±1.03 cups) or years of physical/sports activity respectively and 1) the initial TUG performance duration (rp=-0.167, p=0.568, rs=0.303,p=0.292) nor 2) the gains (%) in session-1 (rp=-0.22, p=0.428, rs= 0.158, p=0.575) nor 3) the gains in session-2 (rp=0.48, p=0.872, rs= 0.485, p=0.079). 

However, the study was not designed (or powered) to test for factors in daily life that may (potentially) affect TUG performance and learning. 

C - l. 494: Why are the results of the transfer test not reported?

R: The transfer test is not reported because reporting the transfer effects (on the order of 98% of the gains in trial duration, already on the initial segment; thus, no large differences from the trained conditions) in full (i.e., kinematics) requires extending the length of the paper and the introduction of a second issue (a point that does not directly relate to the main point of the currents paper). Transfer effects are not the focus of the current paper. 

Given that the transfer tests occurred after the retesting of retention on session-3 we think that the paper presents a full story of how TUG is learned as a novel task that stands by itself.

C - l.494-499: How many participants executed one or more additional trials? I think this needs to be reported. If performing only 18 practice trials even one additional trial could influence the learning curve, and thus makes it difficult to compare the data.

R: We do not think that it is likely that after the attainment of the plateau phase the time course of learning should be critically affected by the addition or subtraction of one or two trials (this is directly indicted in the notion of the plateau phase; please see our note on the pilot protocol results). 

As for additional trials: one subject executed two additional trials (one on session-1 and one on session-2) as replacement trials and one subject executed one additional trial on session-3. 

Given the reviewer’s comment we added the following statement:

“In cases in which a IMU was dislodged or an unexpected distraction occurred (e.g., someone entering the room wherein the test took place), the participants were asked to execute an additional TUG iteration as a replacement. Trials in which the participant made the turn earlier than required were not included in the statistical analysis. One participant executed two additional trials (one on session-1 and one on session-2) as replacement trials and another participant executed one replacement trial on session-3.”

C - l. 533-538: What was the rational for using a SD larger than 3.5 as cutoff. Is this cutoff based on previous experiments?

R: The cutoff value of 3.5 SD was used in determining the initiation (and end point) of the trial in the algorithm for analyzing IMU recordings. This is a very conservative criterion: looking for a deviation on the order of 99.98% of the mean activity during the rest intervals before and after each trial and was found to be sensitive and accurate (using the 0.1 sec sliding window) on a subset of data, before implementation on the complete data set. Nevertheless, all algorithm determinations were also re-viewed and manually corrected: "The criterion for manual corrections, for the two cut-off points, was a flat line (fluctuations of less than 0.1 radian/sec.) in at least one of the axes for at least 0.05 sec. for both the start and the end time-points. The corresponding data from the gyroscope of the IMUs of both legs, in the Z axis (forward-backward direction) was used as an auxiliary cue for determining cut-off points; the leg gyroscopes often provided long segments (>0.2 sec.) of a flat line (fluctuations of less than 0.1 radian/sec.) at the start and end time-points."

The threshold in the current study is slightly more conservative than the threshold (3 SD) used in other studies for delineating the beginning (or end) of activity in a trial (e.g., Frenkel-Toledo, Yamanaka, Friedman, Feldman, & Levin, 2019; Hodges & Bui, 1996). The cutoff used in the current study is conservative given the exploratory multi-session and the (potential and in retrospect the actual) dynamic range of the data across various IMU measurements. 

C - General comment statistical analysis: In general, the description of the statistical analysis is very vague. Many analyses reported in the results section are not mentioned in the statistical analysis section (see comments “results”). Please specify the exact method and data you used for analysis (e.g. mixed ANOVA, between- and within-group variables, etc.). Otherwise, it is impossible to replicate the experiment. Further, did the authors check for normality, variance homogeneity and sphericity of the data where appropriate? What method was used for detecting outliers and how were they treated?

R: Given the reviewer comment we rewrote the statistical analysis sections to better reflect the statistical tests applied to the data sets. For the analysis of a given measure, we first tested for the normal distribution of the data by a Kolmogorov-Smirnov test. For normally distributed data, a repeated measure ANOVA was used with sessions/segments (time-points) as a between-subjects factor and the trials as a within-subject factor which allowed not only a comparison of the mean but also for the testing of the interaction of the two, i.e., the rate of change in the sessions/segments. The results of the repeated measure ANOVAs, were corrected using Huynh-Feldt adjustment, when appropriate, for non-sphericity violation. In order to detect outliers, we calculated for each variable (kinematics) the mean and SD. Variable values above or below 3.5 SDs were considered potential outliers, manually reviewed and excluded from the data set. Non-parametric tests (spearman) were used for variables in which normal distribution was not found. 

 The trial durations in both methods, reaction times, task duration times per trial and the majority of the kinematic measures were normally distributed across all subjects in each trial (excluding mean step lumbar mean acceleration in the x axis, mean step min acceleration in trunk and in the lumbar in the X axis). Unlike per trial durations, the overall (three sessions) trials durations were not normally distributed (presumably resulting from the between-sessions ‘offline’ changes in performance).

In addition, following the reviewer’s comment, in the reporting of statistical results in the Results sections we now specify the nature of the test and the test parameters.

 The revised text for the Methods (statistical analyses) section now reads:

“Statistical analyses were performed both on TUG duration (video, IMUs) and the kinematic measures derived from the IMUs. All statistical analyses were conducted using IBM SPSS Statistics for Windows, Version 25.0 (SPSS Inc. NY, USA). Each variable was first assessed for normal distribution by the Kolmogorov-Smirnov test. Parametric tests, repeated measures ANOVAs and paired, one-sample or independent t-tests and the Pearson correlation test, were used for measures normally distributed measures. Non parametric tests (Spearman) were used for the very few measures indicated by the Kolmogorov-Smirnov test to have a non-normal distribution. 

 The relative changes in task initiation time (RT) across pairs of trials in a session were compared using paired t-tests; the overall gains in session-1 were calculated by the difference from the first to the last trial. Within-session and between-sessions improvements in terms of TUG duration and kinematics, were assessed by comparing four-trial long segments from the beginning and end of each session. Between sessions retention (both overnight and after one week) were assessed by comparing four-trial long segments from the end of one session and the beginning of the following session using repeated measures ANOVAs. To this end, repeated measures ANOVAs with session/segments (time-points) as a between-subject factor and the trials as a within-subject factor. To assess possible differences in the two measures of task execution duration (video, IMUs), a repeated measures ANOVA was used on data from all subjects in all 18 trials of session-1 (trials as a within-subject measure). To evaluate the two phases of performance improvement during session-1, a repeated measures ANOVA was used to compare TUG duration (as a within-subject factor) between the first phase (trials 1-9) and the second phase (trials 10-18) of the session. The Huynh-Feldt adjustments were used, when appropriate, for non-sphericity violation. 

The group mean linear and power trend lines were determined using a Matlab algorithm (Matlab vers. R2019b, The Mathworks, Inc., Natick, MA). The difference between the linear trend line slopes and zero was assessed using one sample t-tests.

 The correlations between TUG duration measurements according to the video and the IMUs and the correlations of the gains in TUG duration and the change in kinematic measures were assessed by either the Pearson or Spearman's correlation methods depending on whether the data distribution was normal or not, respectively. Descriptive statistics of the participants are presented by a mean (±SD), median (minimum-maximum) or number (percentage).”

C - l. 563: Why did the authors use Spearman correlation instead of Pearson correlation to compare TUG duration between video recording and IMU data? If data was not normally distributed why did the authors use ANOVA to analyze data?

R: ANOVAs were only used when the data going into the analysis was normally distributed according to a Kolmogorov-Smirnov test. The TUG durations (in both measures, video and IMUs) per trial across all tested segments and sessions were normally distributed, so a Pearson correlation test was used. However, the Kolmogorov-Smirnov Test showed that (only) the compound data of all the study trials, in all subjects were not normally distributed of (p<0.05) which is why a Spearman test was used. Nevertheless, when comparing the two measurements using the Pearson correlation, there was a high correlation between the two measurement methods (video, IMUs) (r678=0.858, p<0.001). We add the following figure to show the correlation between the two variables. 

Given the reviewer’s comment we now report the Pearson correlation coefficient. 

C - l. 563-570: The authors used only one trial (i.e. first and last trial) to calculate relative improvements in session 1 (i.e. within-session improvement). In contrast, in the next sentence the authors state that they used 4 trials to analyze within- and between session changes. Finally, session 1 was divided in two phases each consisting of 8 trials. Again the number of trials used for analysis seems to be arbitrary. Please specify why you used this specific number of trials for each analysis (e.g. is 1 trial or are 4 trials appropriate to account for motor fluctuations while not masking learning gains).

R: The number of trials in the analysis was chosen according to the aims and hypotheses of the study. Throughout, consistently, all the data were analyzed using 4 trial segments in order to account for motor fluctuations while not masking learning gains. There were two exceptions which are clearly justified. First, to calculate the relative improvement in the first session we referred to the first and last trials due to the fact that the slope of the beginning of the session is the steepest and we wanted to address the entire improvement curve. Second, the 9 trial segment in the 1st session was determined empirically as only in this segment there were trial-by-trial gains; performance in the later trials was at plateau. Thus, the 9 trial segment was an empirical result when addressing the research question relating to the different phases in session-1 as expected in the learning of a new skill (e.g., Karni, 1996; Korman et al., 2003)). 

Throughout the study, to evaluate the differences between and within the sessions and to assess whether the interaction of the two is significant, we used 4 trial segments. The size of segments was determined by the length and structure of the later sessions (2 segments in session-2, 3 segments in session-3). As the reviewer mentioned the 4 trials segment was appropriate to account for motor fluctuations while not masking learning gains. 

C - l. 570-571: I would appreciate if the authors could elaborate on why it is meaningful to correlate offline learning changes with PSQI values in this context. The PSQI retrospectively includes average sleep behavior of the last four weeks. This may not reflect sleep quality of the night between encoding and retention sessions.

R: The question addressed was whether the quality of sleep correlates with the offline changes in behavior given that studies suggest that sleep may be relevant for expressing consolidation phase gains (e.g., Korman et al., 2007, 2003). The PSQI provides a measure of sleep habits during the experiment (note that some offline effects actually emerged between session-2 and session-3). As stated in the methods section we asked participants about their sleep (duration) on session 2. No correlations with performance were found, except for a positive correlation between sleep duration in the night following session-1 and the relative performance gains attained between sessions 1 and 2 (rp=0.637, p=0.014 ,N=14) for the video data but not according to the IMUs (rp=-0.001, p=0.996, N=15).

 Given the reviewer’s comment the text now reads: “Sleep quality, as assessed by the PSQI questionnaire score, did not correlate with the overnight change in performance (Spearman, r=-0.046, p=0.872; r=-0.188, p=0.519; video and IMUs based, respectively) nor with the overall gain across the three sessions (Spearman, r=-0.67, p=0.813; r=0.-0.083, p=0.768, video and IMUs based, respectively).”

Results:

C - In general, the results are clearly structured. 

R: We thank the reviewer for noting that the results are clearly structured. We believe that the revised text (corrected in the light of the following comments) is now even easier to follow and clearly presents how the data were analyzed.

C - However, most of the statistical analyses are not mentioned in the methods sections. This makes it difficult to follow and almost impossible to understand the way how data was analyzed. In my opinion, this is a major issue of the manuscript. 

R: The Methods section (statistical analyses) has now been revised to more comprehensively present the way statistical analyses were done. Please see above.

C - l. 159-162 the authors report the performance improvements in session 1. Statistics indicate that the authors used a repeated-measures ANOVA for analysis including only the first and last trial (e.g., one degree of freedom). It is not clear why they used ANOVA including only two variables instead of paired sample t-test or alternatively why they didn’t incorporate all practice trials of session 1 in the ANOVA.

R: The relative improvement was assessed by the difference from the first to the last trial because as one can see from the data presented in Figure 2 (and as is characteristic of fast learning) there were large gains between the first and second trials. The finding of significant overall gains in the session are provided (i.e., all practice trials of session 1) in the ANOVA comparing the two measurement methods (video, IMUs) (a paragraph above). We also did a first segment (4 trials) to final segment (4 trials) comparison using a rm-ANOVA to highlight the interaction – the rate of change in the two segments was different. This was in accordance with the manner in which within-session changes were assessed across all three sessions (methods). 

 Given the reviewer’s comment we rephrased the result section: “Overall, TUG duration was significantly shortened across the first session (trial 1 to trial 18) in all participants; by 15.0%±5.3 (range: 4.8% - 25.7%) according to the video recordings and by 15.0%±7.7 (range: 2.5% - 35.5%) according to the trunk IMU. A comparison of the first segment (4 trials) of the session to the final segment (4 trials) again showed the significant improvement in TUG duration (F(1,13)=59.602, p<0.01, ŋp2=0.821; F(1,12)=28.52, p<0.01, ŋp2=0.687; video and IMUs, respectively) but also a significant interaction (F(3,39)=5.907, p=0.005, ŋp2=0.282; F(3,39)=8.689, p<0.01, ŋp2=0.401; respectively) indicating that the rate of change across trials decreased significantly across the session (Fig 2). As can be seen in Fig 2 the video and IMUs derived measurements of TUG duration were highly correlated across the three sessions (r678=0.858, p<0.001) and there was no significant difference between the two measures across the 18 trials of the session (F(1,12)=0.275, p=0.61)." 

C - Further examples are: l. 164-166 (in the methods section the authors only mention that video and sensor data are compared using Spearman correlation),

R: See response above to comment of l. 563. Additionally, we rephrased the paragraph: “As can be seen in Fig 2 the video and IMUs derived measurements of TUG duration were highly correlated across the three sessions (r678=0.858, p<0.001) and there was no significant difference, in repeated measure ANOVA, between the two measures across the 18 trials of the session (F(1,12)=0.275, p=0.61).” 

C - l. 167-172 (information on the method and analysis of reaction time is completely missing), 

R: Given the reviewer’s comment we now added in the methods section: “A reaction time measure (RT) was defined as the time to initiate the TUG movement sequence after the 'go' signal (video based).”

In the results section we write: “The gains in TUG duration attained across the session were not due to a shortening of the time to initiate the TUG movements sequence after the 'go' signal, i.e., reaction time (RT) (Fig. 2-inset). In session-1, the overall gains in RT were very small 0.08±0.18 seconds (sec.); there was a small but not significant gain in RT between the first and second TUG trials of the session (paired t-test, t14=1.61, p=0.13) and RTs in trials 2-5 did not differ from RTs in trials 15-18 (F(1,12)=1.696, p=0.217). Moreover, the group mean linear trend line from the 2nd to the 18th trial was y=-0.001x+0.2013; the slopes of the individually fitted linear trend lines, generated for each of the participants, were not significantly different from zero (t14=-1.433, p=0.174)."

C - l. 173-175 & 182-184 (no information on the slope and power function analysis is provided in the methods section), there is no information on the statistical analysis of step kinematics in the methods section, etc.

R: We added to the statistical analysis: “The group mean linear and power trend lines were determined and correlated using a Matlab (Matlab vers. R2019b, The Mathworks, Inc., Natick, MA). The difference between the linear trend line slope and zero was calculated using a one sample t-test.” We added the statistical analysis of the step kinematics – see response above to the ‘general comment’ on the statistical analyses.

C - As aforementioned in methods section, I’m missing information on assumption checks for the statistical tests. Closely related to this comment, did the authors check for normal distribution of the TUG data or adjust for baseline performance? I would assume that performance in such a task could be highly variable between participants due to previous experience.

R: See response above to the ‘general comment’ on the statistical analysis.

C - l. 184-190: I would like if the authors could elaborate on how and why they decided to split the learning curve of session 1 in two phases (i.e. trial 1-9 & 10-18)? Was their any mathematical or statistical rationale? Did the authors check for best model fit (e.g. using different number of trials)?

R: A “split” in session-1 was specifically looked for as two consecutive phases have been suggested to characterize the accruement of performance gains in the initial practice session of a novel task. The two phases were explored in answering one of the main research questions: is the learning in the session showing (following) two phases: fast learning and a plateau phase; rapid gains in performance occur early on in training ("fast learning"), but after a certain number of within-session task iterations, if the task demands remain unchanged, performance levels off ("plateau phase") (e.g., Adi-Japha, Karni, Parnes, Loewenschuss, & Vakil, 2008; Elion et al., 2015; Hauptmann & Karni, 2002; Karni & Sagi, 1993; Korman et al., 2003). Both these phases, in the acquisition of new skills, are well modeled by power functions (Adi-Japha et al., 2008; Blischke & Malangré, 2017; Casabona, Valle, Cavallaro, Castorina, & Cioni, 2018; Elion et al., 2015; Korman et al., 2003).

The decision on how to 'split' session-1 to the different phases was made by using a sliding window analysis on the learning curve for slope changes across trials (e.g., Adi-Japha et al., 2008) and then tested by assessing increasing numbers of trials to best fit the power function model. By splitting session-1 into two parts, the first half best correlated with the power function model and the second half showed a slope not significantly different from zero. As described in the text: "Overall, the gains in group mean TUG duration across session-1 were highly-correlated with a power function model (R2=0.95, 0.94, video and IMUs, respectively) (Fig. 3A). However, two distinct phases of learning were clearly identifiable; an initial phase in which TUG duration became increasingly shorter on consecutive trials (trials 1-9, fast learning), and a second phase wherein performance remained quite stable (trials 10-18, plateau). Trials 1-9 of session-1 were well-fitted by a power function (R2=0.98, 0.97, video and IMUs, respectively). However, the individually fitted linear trend lines for trials 10-18 were characterized by very minimal slopes (y=-0.0033x+6.1154, y=0.0022x+6.118 video and IMUs, respectively; a one-sample t-test showed that the trend line slopes at the latter part of session-1 were not significantly different from zero (t14=-0.335, p=0.742; t14=0.176, p=0.863, video and IMUs, respectively)." 

In order to better explain the reason to 'split' session-1 into two different phases, we rephrased a sentence in the introduction and it now reads: "The aim of the current study was to test whether the performance of common daily activities, included in the TUG test, can be significantly altered (improved) through structured task repetition in young healthy adults; specifically, whether improvements in TUG performance will show fast adaptation or effective transfer of the existing well-honed sub-skills to the particular requirements of the training session, or rather suffice to induce de novo learning, 'fast learning' and plateau and subsequently post-session consolidation phase changes in performance and kinematics."

C - l.190-191: Please rephrase.

R: The sentence now reads: “A repeated measures ANOVA comparing trials 1-9 to trials 10-18, showed a significant difference between the two phases (F(1,11)=69.04, p<0.01, ŋp2=0.863; F(1,11)=32.519, p<0.01, ŋp2=0.747, video and IMUs based TUG duration, respectively), a significant Trial effect (F(8,88)=12.06, p<0.05, ŋp2=0.52; F(8,88)=8.77, p<0.01, ŋp2<0.05) as well as a significant Phase X Trial interaction (F(8,88)=11.24, p<0.01, ŋp2=0.506; F(8,88)=11.23,p <0.01, ŋp2=0.505).” 

C - l. 207-208: Please provide statistics.

R: We added the statistics as recommended and the paragraph is now reads: "There was no significant difference between males (6/15) and females in terms of the magnitude of the TUG duration gains attained in session-1 (independent t-tests, t13=5.589, p=0.904, t13=3.11, p=0.535, video and IMUs based TUG duration, respectively)."

C - I think, it would be helpful for the readers if the authors provide the data of the offline changes (e.g. in percent like they did for online gains).

R: The (offline) gains in terms of the mean step mean acceleration in the x-direction, measured in the trunk and lumbar IMUs, (final segment of session-2 and initial segment of session-3) were on the order of a 6% improvement compared to performance at the end of session-2 (final segment). This was added to the results section.

Discussion:

C - The discussion is clearly structed and good to follow. 

R: We thank the reviewer for stating that the discussion is clearly structured and good to follow.

C - However, I would rather like if the authors could primarily elaborate on why the TUG task, the way the task was administered (e.g. number of practice trials, break duration, feedback, etc.), and the analysis (e.g. splitting session 1 in two phases, number of trials used for analysis, etc.) is appropriate to answer the research question and to draw these conclusions (see my previous comments).

R: We believe that we elaborated this/these points, following the reviewer’s comments on these issues (choosing the TUG task, the research hypotheses, looking for the 2 phases in session-1) in our responses to the comments about the introduction section (see responses re Introduction, methods, data analysis and results). The introduction, methods and results sections were amended to explain these points.

'Figures:

C - I think, including bar charts showing the within- and between session changes (i.e. online and offline changes) would improve the quality of the figures.

R: We considered the reviewer’s suggestion to show the within- and between session changes in a bar chart. We think that as this paper addresses learning in a task composed of everyday activities that have been extensively practiced and executed by all participants during their daily living and that has not been previously considered as a ‘de novo’ learnable task the learning curves should be presented in full. We definitely do that in presenting learning data concerning novel (but also familiar) tasks acquired solely in laboratory settings. We have to show that in this ‘ordinary’ well practiced task, the characteristic phases of new skill acquisition can be delineated. This requires presenting the lab practice related changes in performance in a trial by trial manner. 

Reviewer #2: 

In this study, the authors investigated the short-term and long-term effects of repeated executions of the Timed-Up-and-Go (TUG) task (a clinical test of daily function), a task comprised from a sequence of motor actions: rising from a seated position, walking, turning, and returning to a seated position. The study participants (N=15) were young healthy adults who practiced the TUG task on the 1st day 18 times and retested 24 hours and 1 week post-practice. Performance characteristics were derived from video recordings and 5 wearables with gyroscope and accelerometers. The results showed that the duration of TUG execution was shortened, and that the trial-to-trial changes were fitted by a power function. Further off-line gains in performance were obtained in the subsequent test-sessions. Different kinematic parameters followed different time-course of practice-related changes: some were related to within-session, while other to between-session changes. The authors conclude that their results show that even common everyday motor tasks can be improved by practice.

This is an interesting manuscript with many strengths, it supplies novel, clinically relevant, and theoretically important results. Specifically, the study adds new knowledge to the theories of skill acquisition and maintenance, generally assumed to be relevant mainly to the learning of new motor skills and/or task conditions. The manuscript presents sound methodology and analysis, however the theoretical basis (the structure of the literature review) and the rational/hypotheses are not clear enough. Therefore, the manuscript needs several clarifications and revisions before I can endorse the publication. My points are listed below.

Response to Reviewer #2:

We thank the reviewer for the supporting comments. We appreciate the reviewer’s comment that our is “an interesting manuscript with many strengths, it supplies novel, clinically relevant, and theoretically important results. Specifically, the study adds new knowledge to the theories of skill acquisition and maintenance, generally assumed to be relevant mainly to the learning of new motor skills and/or task conditions. The manuscript presents sound methodology and analysis.” 

We have carefully considered the comments and suggestions made by the reviewer and are confident that the revised manuscript now addresses these points and by incorporating in the revised manuscript additional information and clarifications as suggested by the reviewer, the clarity of our paper is improved. 

In the following we explain how we addressed the major points raised in the reviewer’s comments and suggestions in a point-by-point manner.

Technical points:

1. Please, incorporate figures and legends in the text. The old-style arrangement of the figures at the end of the document is very inconvenient for reading from PDF (as opposed to printed manuscript).

1Response: We appreciate the reviewer's comment about incorporating the figures and legends in the text. However, the submission guidelines of PLOS ONE journal state: "Do not include figures in the main manuscript file. Each figure must be prepared and submitted as an individual file. Figure captions must be inserted in the text of the manuscript, immediately following the paragraph in which the figure is first cited (read order). Do not include captions as part of the figure files themselves or submit them in a separate document."

 2. The resolution of the Figures is very low, please, improve their quality. In the current version the axis names are barely seen.

2R: We thank the reviewer for the comment. In the revised version we improved the quality of all the graphs, and we made sure that the figure’s resolution is higher.

3. The style and the size of the font are inconsistent. Please, check.

3R: Thank you. We have checked the style and the font size to ensure consistency throughout the manuscript. 

Major points:

4. Line 65: I would expect here a sentence about deliberate practice models, as they refer to long-term improvements in non-novel tasks.

4R: Given the reviewer’s comment we have added to the Introduction: " However, we do not expect that the characteristic phases of new learning will replicate in a task which is already over-trained, as in walking or typing. In other words, at issue is the question of whether we should expect to see the characteristic time-course of new skill acquisition (rather than fast adaptation (Krakauer et al., 2019; Shadmehr et al., 2010; Telgen et al., 2014), and effective transfer (Rosalie & Müller, 2014) occurring in well-rehearsed (familiar) tasks such as, for example, typing when we introduce a new text or raising from an unfamiliar chair. Viewed from a related framework there is the notion that very extensive, long-term and deliberate practice is required to gain, albeit via incremental gains, in well rehearsed tasks in order to attain expertise in different domains (Ericsson et al., 1993). Thus the question would be whether if afforded an opportunity for deliberate practice, young adults would show a pattern of continued incremental improvement or whether the deliberate-structured practice opportunity would result in a form of ‘de novo’ learning for a highly practiced, everyday task (Baker & Young, 2014; Ericsson & Harwell, 2019; Krakauer et al., 2019; Yang et al., 2021)."

5. Lines 83-95: The general message of the paragraph is unclear. Is it about laboratory vs. ecological settings, about generalization of principles of learning simple tasks vs. complex tasks or about the determinants of long-term motor task presentation? Balance tasks are not necessarily more complex than sequence task. I suggest avoiding “complexity” issues or providing a clear operational definition, if complexity is a key construct the authors are interested to discuss. In my view, a simple taxonomy, sequence vs. adaptation motor tasks is enough.

5R: We thank the reviewer for the suggestion of changing the taxonomy. One can address repeated TUG performance as practicing a sequence or an adaptation task (e.g., chair height, path structure) or both. We chose to use the designation of "complex" task after reviewing different definitions. Early definition described task complexity as the demands and memory processing imposed in each sub-task on the learner (Naylor & Briggs, 1963). Later, task complexity was defined based on the number of movement segments (Fontana, Furtado, Mazzardo, & Gallagher, 2009; Magill, 2000) or part-tasks (Roessingh, Kappers, & Koenderink, 2002). One review judged a task to be complex if it cannot be mastered in a single session, has multiple degrees of freedom, and is ecologically valid (Wulf & Shea, 2002). Task complexity was additionally defined by the interaction among levels of expertise, number of task parts, and cognitive ability (Fontana et al., 2009). In a recent review, tasks that were describe in the literature as "complex" are characteristically composed of multiple sub-tasks, require control of multiple body movement segments, with multiple degrees of freedom, and involve a rich dynamic interaction with the environment (Du, Krakauer, & Haith, 2022).

However, given the reviewer’s comment we decided to avoid the taxonomy of simple vs complex motor task and we rephrased the paragraph in the following manner: 

"The time-course of performance changes in more common everyday motor routines, such as walking or upright standing, have been less studied (Al-Sharman & Siengsukon, 2013; Elion et al., 2015; Freitas et al., 2020) and when studied, these tasks were followed in special laboratory conditions, i.e., not in everyday or well-rehearsed conditions. The generalizability of the results from studies using laboratory tasks to the learning of real world motor skills has been questioned (Krakauer et al., 2019; Wulf & Shea, 2002). Some evidence, however, does suggest that there is a correspondence between the time-course of learning and skill consolidation in the acquisition of voluntary simple manual skills (e.g., FOS or finger tapping) and the time-course of learning and skill consolidation in gross motor skill e.g., balance maintenance tasks,(Casabona et al., 2018; Elion et al., 2015; Valle et al., 2015). Also, as in the FOS learning task (Korman et al., 2007; Walker et al., 2002), sleep can facilitate off-line gains tasks such as walking (an irregular elliptical path) while doing mental arithmetic (Al-Sharman & Siengsukon, 2013); although the type of task and training parameters may moderate the necessity of sleep (Christova et al., 2018)." 

 6. Lines 92-95: I find it necessary to include the type of instruction and/or feedback as important determinants of the course of sequence learning (e.g., self-paced, cue-triggered performance sequence-of-movements / with or without knowledge of results).

6R: Given the reviewer’s comment we rephrased the text addressing the trial instruction and the feedback (not) afforded to the participants in the task: 

"In the current study, participants were instructed to walk as fast as they can, but without running; it was stressed that while speed was essential, running was to be avoided. Before each trial, the participants were asked to lean against the backrest until they hear the cue to start; to keep their hands crossed on the front of their chest throughout the trial; to walk in the instructed path as fast as they can but without running; when returning to the chair and re-seating, to lean against the backrest until they hear the cue that the trial is finished. Participants were asked if they were ready to continue before the initiation of the next trial and instructed not to speak during the trial. The participants did not receive any information about their actual performance during sessions, except for being cautioned not to run, but were given general encouragement in the form of “OK/good, when you are ready let’s do it once more”.” 

7. Lines 103-110: Wouldn’t it be more precise to use the term “motor actions” instead of “activities”?

7R: We think that the term ‘activities’ is the more appropriate here. Daily motor actions can suggest that these meaningful activities are the product of the motor system by itself rather than activities as context dependent task solutions for daily tasks and situations. 

8. Lines 107-126: What is surprising here? Classical paradigms of sequence learning are based on practicing a specific order of finger/arm movements. The basic elements, e.g., finger-to-keyboard movement, are well-trained by definition. The TUG task also focuses on a specific sequence of well-trained (this time, whole-body) motor actions with novel task demands (perform as fast as possible, but don’t run). As well, the TUG task demands to keep the hands on the chest while walking – an uncommon motor constraint that can’t be considered a “highly familiar sub-task and routines”. Please, clarify this point in the introduction and in the discussion.

8R: The reviewer raises the question of the novelty of this study and specifically why learning curves of whole-body sequence motor tasks should be different from typical laboratory motor tasks if both are classical paradigms of sequence learning. We do not think that there is a reason for assuming that learning in real life situations and in the lab would differ in a fundamental way (with some caveats concerning the relative complexity of tasks, as the reviewer mentioned); however, there is a very different pattern of performance changes across task iterations (practice effects) in novel tasks (“de novo” learning) and in well-rehearsed tasks. 

The difference is clear from studies of laboratory tasks across multiple sessions: the learning curves one finds in the early sessions, specifically the 1st learning-practice session (in both perceptual and motor tasks), markedly differs from the learning curves one finds in later sessions. Indeed, it has been suggested that the characteristics of the learning curves across task repetitions can be used as indicators of whether a task is novel or a previously well-trained one (novelty effect; e.g., Korman, Raz, Flash, & Karni, 2003).

In the real world, we do not expect that being requested to walk quickly in a corridor in a lab (with no obstacles) would show gains in performance characterizing the learning of a new task or cause significant changes in the person’s subsequent walking routines, because a highly trained routine, as walking, is expected to show ‘transfer’ and, if at all, some small adaptation gains between the first and the second or perhaps even the second and third trials rather than to show a “fast learning” phase as in the performance of a novel task (e.g., Korman et al., 2003). We expect ‘expert’ walkers to walk in a corridor using a set of well-established routines, even with the added constraint of crossing one’s hands on one’s chest. 

Laboratory tasks, both perceptual and motor, show consistently that the within-session improvements in performance that occur across repeated task iterations minimize or disappear beyond the second or third training sessions. In fact, in laboratory setting, there are no significant within-session gains after the tasks has been trained for a few sessions (except small readaptation gains between first and second iterations in the later sessions) (Gal et al., 2019; Karni & Sagi, 1993; Korman et al., 2007, 2003).

 So the questions addressed are whether we should expect to see 'fast (de novo) learning' occurring in well-rehearsed (familiar) tasks (rather than fast adaptation (Krakauer et al., 2019; Shadmehr et al., 2010; Yang et al., 2021) or effective transfer (Rosalie & Müller, 2014)? Second, can we see evidence for consolidation phase (post-session) ‘delayed’ gains (Korman et al., 2003; Telgen et al., 2014)? For example, would one expect a professional basketball team to undergo a new learning process (relating to ball shooting) each time they switch to a new court? In our current study, we show not only performance improvement (task execution times) but changes (that persist and build up beyond the session, i.e., a pattern of delayed gains) in the kinematics of the walking segment of the TUG, in young healthy participants (for theoretical perspectives see (Krakauer et al., 2019; Yang et al., 2021). 

It has been suggested that different neuronal mechanisms underlie the recalibration of an existing versus acquisition of a new task performance controller mechanism and the generation of a new performance routine (‘de novo’ learning) (Karni, 1996; Krakauer et al., 2019). A fast learning phase in the initial learning session (Karni & Sagi, 1993; Korman et al., 2003), a between-sessions shift in speed-accuracy tradeoff (Korman et al., 2003; Telgen et al., 2014) and subsequently long-term, “offline”, gains between sessions (Karni & Sagi, 1993; Korman et al., 2003; Telgen et al., 2014) have been suggested as characteristic of the ‘de novo’ building of a task specific controller and the generation of a new performance routine. 

 Given the reviewer’s comment we have extended the following explanation of the issue in the introduction: “Because the characteristic phases in the mastering of new tasks can be delineated in, for example, perceptual skills as well (Karni & Sagi, 1993) it has been proposed that the acquisition of a new skill (i.e., the generation of long-lasting practice-dependent gains in ability) reflects a common repertoire of basic neural processes, albeit in the different brain systems that subserve the execution of the different tasks (e.g.,Karni, 1996; Karni et al., 1998). Thus, the time-course of skill acquisition rather than the specifics of the task may best reflect processes of neuronal and systems level brain plasticity (Dudai et al., 2015; Korman et al., 2003; Telgen et al., 2014). This would suggest that if young adults do acquire new or additional skill when afforded the opportunity to practice common everyday activities, the time-course of performance improvement would follow that of new skill learning in laboratory tasks and conditions. However, we do not expect that the characteristic phases of new learning will replicate in a task which is already over-trained, as in walking or typing. In other words, at issue is the question of whether we should expect to see the characteristic time-course of new skill acquisition (rather than fast adaptation (Krakauer et al., 2019; Shadmehr et al., 2010; Telgen et al., 2014), and effective transfer (Rosalie & Müller, 2014) occurring in well-rehearsed (familiar) tasks such as, for example, typing when we introduce a new text or raising from an unfamiliar chair. Viewed from a related framework there is the notion that very extensive, long-term and deliberate practice is required to gain, albeit via incremental gains, in well rehearsed tasks in order to attain expertise in different domains (Ericsson et al., 1993). Thus the question would be whether if afforded an opportunity for deliberate practice, young adults would show a pattern of continued incremental improvement or whether the deliberate-structured practice opportunity would result in a form of ‘de novo’ learning albeit for a previously highly practiced, everyday task (Baker & Young, 2014; Ericsson & Harwell, 2019; Krakauer et al., 2019; Yang et al., 2021)."

 In addition, we return to this point in the discussion as well: 

"Importantly, learning and retention during, and following, TUG practice proceeded in the distinct time-course of learning and retention previously described in the acquisition of new finger movement tasks (e.g.,Korman et al., 2003), as well as when participants practiced gross-body movements in novel, unfamiliar, laboratory conditions (e.g., Elion et al., 2015). Thus, in the well-rehearsed (familiar) sequence of rising-walking-sitting composing the TUG task, the gains followed the characteristic time-course of de novo learning, 'fast learning' and plateau, rather than fast adaptation (Krakauer et al., 2019; Shadmehr et al., 2010) or effective transfer (Rosalie & Müller, 2014). Moreover there is evidence for a (post-session) consolidation phase in the form of ‘delayed’ (and well retained) changes in kinematics in the walking segment of the TUG(Korman et al., 2003; Telgen et al., 2014)."

 9. Lines 127-132: Please, include specific hypotheses of the study.

9R: We rephrased our hypotheses in the introduction: "The aim of the current study was to test whether the performance of common daily activities, included in the TUG test, can be significantly altered (improved) through structured task repetition in young healthy adults; specifically, whether improvements in TUG performance will show fast adaptation or effective transfer of the existing well-honed sub-skills to the particular requirements of the training session, or rather suffice to induce de novo learning, 'fast learning' and plateau and subsequently post-session consolidation phase changes in performance and kinematics. Our work hypothesis was that affording young healthy adults an opportunity to practice (repeat in a deliberate-structured manner) a sequence of common, everyday, highly trained sub-skills, may suffice for initiating ‘de novo’ skill learning. We show that TUG practice induced robust and enduring gains in performance as well as robust and enduring changes in the kinematics of the everyday activities when performed in the context of the practiced task.” 

 10. In the Methods, please, include justification for the sample size.

10R: Given the reviewer’s comment we now include our justification for the sample size in the participants section: 

“Group size was determined based on the temporal data from a pilot study, using i. video-recordings analysis and ii. data derived from the IMUs. The mean and standard deviations of the initial and final four trials of the 1st session, and of the first four trials in session-1 and session-2, were used to assess effect sizes. We calculated the sample size with a) the Simple Interactive Statistical Analysis (SISA) online software (http://www.quantitativeskills.com/sisa/) and b) with the ClinCalc calculator (www.clinicalc.com). According to both calculators, the number of participants needed was 2 – 22 (median 7-10) depending on the measure used (video, IMUs). Note however that these numbers were derived for comparing across groups; the main comparisons in this study were within-subjects, necessitating smaller numbers of participants.”

Thus, a sample size of 15 participants was deemed reasonable.

11. In the Methods, what does it mean: “The participants were partly reimbursed for taking part in the study.”?

11R: The sentence was rephrased: “The participants were paid a small set sum for participation in each session.” 

12. In the Methods, please, clearly state what type of feedback was provided to the participants.

12R: In the revised Methods section we now write: "Participants did not receive any information about their actual performance during sessions, except for being cautioned not to run, but were given general encouragement in the form of “OK/good, when you are ready let’s do it once more."

13. It would be helpful to have a video recording of a representative trial presented in the supplementary information.

13R: We will attach a representative trial in the supplementary information; the participant gave his consent for this publication. 

14. Lines 173-175: “with the linear trend line slopes of the participants” – please, rephrase. The slopes are individually fitted to the RTs across trials for each participant, I guess.

14R: We rephrased the sentence as suggested by the reviewer: “Moreover, the group mean linear trend line from the 2nd to the 18th trial was y=-0.001x+0.2013; the slopes of the individually fitted linear trend lines, generated for each of the participants, were not significantly different from zero (t14=-1.433, p=0.174).”

15. Line 222: Why the authors substitute “TUG duration” by “speed”? Please, justify or use a consistent name for the dependent variable.

15R: We rephrased all references to "speed" to “TUG duration".

16. Lines 231-236: As transfer tests are excluded from the current paper, but could affect the performance of the trained TUG direction, I suggest excluding the report and discussion of additional gains in TUG duration accrued across session-3. Please, present the per cent of improvement from the baseline to the initial, pre-transfer test trials of session-3.

16R: As the length of session-1 and session-3 are similar (18 and 20 iterations respectively), it is important, to our mind, to show that in session-3 there were additional gains between the first and last trials segments as evidence that the plateau phase in the second half of session-1 was not due to fatigue or a general floor effect. However, given the reviewer’s comment we added the percent of improvement from the first trial to the last trial of the pre-transfer segment of session-3 in addition to the overall improvement. 

The results section was rephrased given the reviewer’s comment: "The gains in TUG duration attained by the end of session-2 were well-retained by session-3, i.e., over a one-week interval (F(1,14)=1.727, p=0.210). Moreover, additional gains in TUG duration were accrued across session-3, with faster performance in trials 5-8 compared to trials 1-4 of the session (F(1,14)=6.626, p=0.022, ŋp2=0.321); TUG duration continue to improve in trials 17-20 compared to trials 5-8 of the session (F(1,14)=6.289, p=0.025, , ŋp2=0.310). Thus, overall, the gains in TUG duration were on average 15.08% (range: 6%-30%) and 18.0% (range: 0%-37.3%) across the three sessions, computed to before and after the transfer test segment, respectively."

We added the following to the supplementary results according to the video-based measurements: " .. The gains in TUG duration attained by the end of session-2 were well-retained by session-3, i.e., over a one-week interval (F(1,13)=0.708, p=0.415). No additional gains in TUG duration accrued between the first two segments of session-3 (F(1,14)=2.162, p=0.164), however, additional gains in TUG duration were accrued after the transfer test segment in session-3, with faster performance in trials 17-20 compared to trials 1-4 of the session (F(1,14)=13.85, p=0.002, ŋp2=0.497). Overall, the average improvement in TUG duration across the three sessions was 19% (range: 3.3%-28.5%); the average improvement attained before the transfer segment of session-3 was 16.54% (range: 5%-24%)."

17. Lines 317-326: Please discuss considering the existing literature in sequence learning referring to the distinct within-session and between-session changes in kinematic characteristics of individual movements, - where different kinematic characteristics show a distinct time course and a specific susceptibility to interference.

17R: Given the reviewer’s comment we added to the discussion some references to the existing literature referring to the distinct within and between session changes in kinematic characteristics in sequence learning. 

"The notion that during the post-session consolidation phase “slow” qualitative changes occur in the trained task’s performance, and the neural machinery serving the execution of the practiced motor task, is suggested by the finding of reductions in performance variability and/or an improved speed-accuracy relationship across the post-session interval (when no additional practice was afforded) (Korman et al., 2003; Shmuelof, Krakauer, & Mazzoni, 2012). Qualitative changes in the movements’ path and in the velocity profiles, indeed the generation of new movement primitives, have also been suggested as the products of consolidation processes(Sosnik, Hauptmann, Karni, & Flash, 2004). A recent study showed that practicing a sequence of locomotor movements resulted in reduced latencies of step initiation and higher peak velocities during the swing phase (Johannsen et al., 2022). The process of improvement does not necessarily rely only on biomechanical optimization; sequence planning, chunking (Panzer & Shea, 2008; Rozanov, Keren, & Karni, 2010; Sosnik et al., 2004) and the within-sequence timing of movements, i.e., the temporal structure of task execution, can undergo qualitative changes as well (Friedman & Korman, 2016; Gal et al., 2019; Orban et al., 2011)."

---

## [Decision Letter · Decision Letter 1]

28 Feb 2023

PONE-D-22-15151R1Improving old tricks as new: young adults learn from repeating everyday activitiesPLOS ONE

Dear Dr. Leizerowitz,

Thank you for submitting your manuscript to PLOS ONE. After careful consideration, we feel that it has merit but does not fully meet PLOS ONE’s publication criteria as it currently stands. Therefore, we invite you to submit a revised version of the manuscript that addresses the points raised during the review process.

We look forward to receiving your revised manuscript.

Kind regards,

Peter Andreas Federolf

Academic Editor

PLOS ONE

Journal Requirements:

Additional Editor Comments:

Please consider the remaining comments of reviewer 1.

Reviewers' comments:

Reviewer's Responses to Questions

**Comments to the Author**

1. If the authors have adequately addressed your comments raised in a previous round of review and you feel that this manuscript is now acceptable for publication, you may indicate that here to bypass the “Comments to the Author” section, enter your conflict of interest statement in the “Confidential to Editor” section, and submit your "Accept" recommendation.

Reviewer #1: (No Response)

Reviewer #2: All comments have been addressed

2. Is the manuscript technically sound, and do the data support the conclusions?

Reviewer #1: Yes

Reviewer #2: Yes

3. Has the statistical analysis been performed appropriately and rigorously? 

Reviewer #1: Yes

Reviewer #2: Yes

4. Have the authors made all data underlying the findings in their manuscript fully available?

Reviewer #1: Yes

Reviewer #2: Yes

5. Is the manuscript presented in an intelligible fashion and written in standard English?

Reviewer #1: Yes

Reviewer #2: Yes

6. Review Comments to the Author

Reviewer #1: Once again, I would like to thank the authors for this interesting study. The authors made large efforts in revising their manuscript. Most of my comments were adequately addressed in the revised version of the manuscript and I think the manuscript substantially improved. Congratulations on this great work. I have only some minor comments, which the authors may want to address:

Method:

• Please include a statement on how participants were recruited.

• l. 662-664: Statistical methods are not mentioned for the within-session analysis. Please add.

Results/Discussion:

• Participant awareness of study aims: The authors mentioned that the participants were aware of the aim of the study. Participants may behave differently when they know the research hypothesis. Could it be possible that the participants behaved/performed differently because they knew that the general aim of the study was to “test whether new learning occurs in everyday tasks”. If a participants assumes that learning could occur this might affect performance. I would appreciate if the authors could elaborate on this and at least mention this in the discussion/limitations of the paper.

• l. 523-526: The authors mentioned that the participants did not receive feedback on their task performance (i.e. knowledge of result). However, it has been shown that knowledge of result can be fundamental for motor learning. I would appreciate if the authors could elaborate on this and possibly add this to their discussion.

• l. 527-529 – breaks between trials: It seems that the break duration considerably differed between participants (15 sec. – 19 min). Research has shown that break duration can greatly impact motor learning. Recent studies even suggest that micro-offline learning may occur during short breaks (Bönstrup et al., 2019, 2020). Even though the authors did not compare different groups the variance/heterogeneity in break duration may have influenced the results of the study. Thus, I would appreciate if the authors clearly highlight this in their paper (e.g. limitations). Did the authors analyze if participants’ performance/learning gains systematically differed as a function of break duration? This might be an interesting finding.

• l. 586-591 – additional trials in some participants: Did the authors check if the data of the participant performing more trials differed from the group average. Even though an additional trial during the plateau phase does not influence performance at the end of practice it could influence consolidation (i.e. offline learning).

• l. 191 & 223: please rephrase statistics.

• I would appreciate if the authors report effect size (e.g. ŋp2, Cohen's d) also for non-significant results.

• l. 259: Please remove quotation mark.

• l. 265: Please remove the second semicolon.

• l. 265-269: It is not clear which trials from which session were analyzed here. Please rephrase.

• I would appreciate if the authors would also report normalized offline learning of the TUG duration (i.e. percent change from end of practice to beginning of retention test) between session 1, 2, and 3 (similar to the reporting of online gains).

Figure:

• I agree with the authors that the trial by trial performance should be reported. However, I think that including additional bar charts showing (normalized) on- and offline learning may improve the results.

• If the data of the transfer test are not reported in the paper I would suggest to remove them from figure 2.

Reviewer #2: The authors comprehensively addressed all my concerns and I am happy to endorse the publication of this interesting and important article

7. PLOS authors have the option to publish the peer review history of their article (what does this mean?). If published, this will include your full peer review and any attached files.

Reviewer #1: No

Reviewer #2: No

---

## [Author Response · Author response to Decision Letter 1]

6 Apr 2023

Reviewer #1: 

• “Once again, I would like to thank the authors for this interesting study. The authors made large efforts in revising their manuscript. Most of my comments were adequately addressed in the revised version of the manuscript and I think the manuscript substantially improved. Congratulations on this great work. I have only some minor comments, which the authors may want to address.”

Response - We thank the reviewer for this comment. We have carefully considered the reviewer’s remaining minor comments and suggestions and hope that the revised manuscript now addresses these points in the revised manuscript. In the following we explain how we addressed the points raised in your specific comments in a point-by-point manner.

Method:

• Please include a statement on how participants were recruited.

Response – We added the following statement to the methods: "The participants were recruited via social media; most participants were university students or staff members at the Sheba Medical Center."

• l. 662-664: Statistical methods are not mentioned for the within-session analysis. Please add.

Response – The tests were repeated measures ANOVAs. The text now reads: "Within-session and between-sessions improvements, in terms of TUG duration and kinematics, were assessed by comparing four-trial long segments from the beginning and end of each session. To this end, repeated measures ANOVAs were used with session/segments (time-points) and the trials comprising them as within-subject factors. Within-session gains were assessed by comparing the four-trial long segments from the beginning and end of each session. Between-sessions changes (both overnight and after one week) were assessed by comparing the four-trial long segments from the end of one session and the beginning of the following session."

Results/Discussion:

• Participant awareness of study aims: The authors mentioned that the participants were aware of the aim of the study. Participants may behave differently when they know the research hypothesis. Could it be possible that the participants behaved/performed differently because they knew that the general aim of the study was to “test whether new learning occurs in everyday tasks”. If a participant assumes that learning could occur this might affect performance. I would appreciate if the authors could elaborate on this and at least mention this in the discussion/limitations of the paper.

Response - Participants were aware to the aim of the study. They were asked to sign a consent form in order to participate in this study which included the following:

"The purpose of the study is to examine the acquisition and learning of a daily motor skill: getting up from sitting, walking around and returning to sitting position (using a test from the rehabilitation world)."

"The General rational and the aims of the study: Repeated practice can lead to learning and improvement of performance. In the present research we will examine whether and how repeating simple everyday activities affects performance. Performance will be assessed by following movement speed and the characteristics of the movement." 

 Throughout the three sessions participants were given the same instructions: to execute the TUG test as fast as they can. They were not asked in each trial to achieve better results than in the previous trial. Participants were not made aware of the fact that we are looking into whether gains in performance follow in different phases. Nor were they aware of which of the kinematic measures will be examined. We cannot assume that knowing the aim of the study was central to modifying the participants’ behavior in a specific manner, as found. 

 In teaching new laboratory motor skill tasks, the participants know the aim of the study and in all tasks what needs to be executed is explicitly stipulated. In most cases participants are also requested to perform as fast and as accurately as possible (Karni et al., 1995). Yet, despite the explicit instructions given to the participants, there is an implicit process of learning of a skill. 

 Given this comment, and further comments (l. 523-526, l. 527-529) made by the reviewer, we added a short statement: "One should note that participants were made aware of the general aim of study when enrolling in the study, and that the task elements and sequence were explicitly stipulated at the beginning of each session. However, throughout the three sessions participants were given the same instructions - to execute the TUG test as fast as they can without running; participants were not asked to achieve better results than in a previous trial. The participants were not aware that we were looking into whether gains in performance would follow in different phases; nor were they aware of which of the task elements or kinematic measures will be examined. The question of whether knowing the aim of the study was central to modifying the participants’ behavior or that similar learning would occur in a more implicit or incidental learning situation is open for further experimentation. Future studies can also address a possible effect of affording feedback to the participants (e.g., Kilduski & Rice, 2003; Wulf, Shea, & Lewthwaite, 2010), the interaction with the teacher-instructor (e.g., Awad-Igbaria, Maaravi-Hesseg, Admon, & Karni, 2022) or the effects of the length of breaks afforded between the trials (e.g.,Bönstrup, Iturrate, Hebart, Censor, & Cohen, 2020)."

• l. 523-526: The authors mentioned that the participants did not receive feedback on their task performance (i.e., knowledge of result). However, it has been shown that knowledge of result can be fundamental for motor learning. I would appreciate if the authors could elaborate on this and possibly add this to their discussion.

Response – We agree. Feedback was found to enhance the learning of a motor skill (Wulf & Lewthwaite, 2016; Wulf et al., 2010) By withholding feedback about the subjects' actual trial-by-trial performance we may have underestimated the potential for task relevant performance gains. 

In the new paragraph about possible caveats we enumerate this point: "Future studies can also address a possible effect of affording feedback to the participants (e.g., Kilduski & Rice, 2003; Wulf, Shea, & Lewthwaite, 2010), the interaction with the teacher-instructor (e.g., Awad-Igbaria, Maaravi-Hesseg, Admon, & Karni, 2022) or the effects of the length of breaks afforded between the trials (e.g.,Bönstrup, Iturrate, Hebart, Censor, & Cohen, 2020)." 

• l. 527-529 – breaks between trials: It seems that the break duration considerably differed between participants (15 sec. – 19 min). Research has shown that break duration can greatly impact motor learning. Recent studies even suggest that micro-offline learning may occur during short breaks (Bönstrup et al., 2019, 2020). Even though the authors did not compare different groups the variance/heterogeneity in break duration may have influenced the results of the study. Thus, I would appreciate if the authors clearly highlight this in their paper (e.g. limitations). Did the authors analyze if participants’ performance/learning gains systematically differed as a function of break duration? This might be an interesting finding.

Response – We report that "The median time between the beginning of one trial and the beginning of the next one was 23 sec. (15 sec. – 19 min, an instance of a IMU problem). The mean break duration was 32 ± 53 sec. (SD)". Both median and average of all the trials breaks were on the order of a half of a minute despite the fact that one break lasted 19 minutes due to technical problems. The difference in break duration was not between participants but rather sparse random occurrences. Only 38 out of 646 breaks [~5%] were longer than one minute. Due to the small number of longer breaks, we cannot perform an analysis on whether participants’ performance/learning gains were affected as a function of break duration. Moreover, the 19 min break did not occur in session-1, it occurred in session-3, well after performance leveled off. However, taking into account the reviewer suggestion, we examined the correlation between the length of session-1 in seconds and 1) the relative improvement in performance in session-1 and 2) the overnight relative improvement in performance. No significant correlation was found (according to either the IMU's or the videos).

We acknowledge the possibility that the length of the breaks between the trials can affect the learning procedure. Recent papers (Bönstrup et al., 2020, 2019) suggested that in early skill learning, short period of rest (on the order of seconds) may result in consolidation. In a follow up study, yet to be published, the participants performed only five iterations of the TUG in the first two sessions; i.e., did not reach the plateau phase within the first session. However, despite the 24 hours break between the first and second session, the learning process continued in the same manner as it had in the presented study. A break of 24hours was not different in terms of TUG duration gains from a ~half a minute long break. 

 Given the reviewer’s comment, we rephrased the methods section and it now reads:" After completing each trial, data were transferred from IMU-level storage and the participants were asked, and indicated, when they were ready for the next trial. The between-trial breaks were on the order of a half a minute (32 ± 29 sec., mean, SD) with one exception - a 19 min break due to technical problem with the IMU." 

Furthermore, we added the possibility to the new caveats-potential future studies paragraph: "Future studies can also address a possible effect of affording feedback to the participants (e.g., Kilduski & Rice, 2003; Wulf, Shea, & Lewthwaite, 2010), the interaction with the teacher-instructor (e.g., Awad-Igbaria, Maaravi-Hesseg, Admon, & Karni, 2022) or the effects of the length of breaks afforded between the trials (e.g.,Bönstrup, Iturrate, Hebart, Censor, & Cohen, 2020)."

• l. 586-591 – additional trials in some participants: Did the authors check if the data of the participant performing more trials differed from the group average. Even though an additional trial during the plateau phase does not influence performance at the end of practice it could influence consolidation (i.e. offline learning).

Response - Only two participants were given additional trials: "One participant executed two additional trials (one on session-1 and one on session-2) as replacement trials and another participant executed one replacement trial on session-3". The participants’ gains across the 3 sessions were within 1.2 and 1 SD of the group mean gains. 

• l. 191 & 223: please rephrase statistics.

Response - The statistics were rephrased in the following manner: 

L. 191: “As can be seen in Fig 2 the video and IMUs derived measurements of TUG duration were highly correlated across the three sessions (rp=0.858, p<0.001, N=678) and there was no significant difference between the two measures across the 18 trials of the session (F(1,12)=0.275, p=0.61, ŋp2=0.022).”

L. 223: "A repeated measures ANOVA comparing trials 1-9 to trials 10-18, showed a significant difference between the two phases (F(1,11)=69.04, p<0.001, ŋp2=0.863; F(1,11)=32.519, p<0.001, ŋp2=0.747, video and IMU based TUG duration, respectively), a significant Trial effect (F(8,88)=12.06, p<0.001, ŋp2=0.52; F(8,88)=8.77, p<0.001, ŋp2=0.424) as well as a significant Phase X Trial interaction (F(8,88)=11.24, p<0.001, ŋp2=0.506; F(8,88)=11.23,p <0.001, ŋp2=0.505).”

• I would appreciate if the authors report effect size (e.g. ŋp2, Cohen's d) also for non-significant results.

Response – There are ongoing debates about what can be inferred from effect size measures when the effect is not statistically significant (e.g., https://www.researchgate.net/post/Effect-sizes-for-non-significant-results). Given the reviewer’s request we added the effect size also for non-significant results. 

• l. 259: Please remove quotation mark.

Response – The quotation mark was removed. 

• l. 265: Please remove the second semicolon.

Response - We have removed the second semicolon from the sentence.

• l. 265-269: It is not clear which trials from which session were analyzed here. Please rephrase.

Response – We thank the reviewer for the opportunity to clarity of the paragraph. The text now read: "Moreover, additional gains in TUG duration were accrued across session-3, with faster performance in trials 5-8 compared to trials 1-4 of the session (F(1,14)=6.626, p=0.022, ŋp2=0.321); TUG duration continued to improve in trials 17-20 compared to trials 5-8 of the session (F(1,14)=6.289, p=0.025, ŋp2=0.310). Thus, overall, the gains in TUG duration across the three sessions, were on average 15.08% (range: 6%-30%) and 18.0% (range: 0%-37.3%) computed to before and after the transfer test segment, respectively."

• I would appreciate if the authors would also report normalized offline learning of the TUG duration (i.e. percent change from end of practice to beginning of retention test) between session 1, 2, and 3 (similar to the reporting of online gains).

Response – Given the reviewer’s comment we added the percent change from end of practice to beginning of retention test across the two intervals between sessions 1-3. In the methods section: "The offline, between sessions, gains in TUG duration were computed as percent change differences between the final trial of sessions 1 and 2 and the corresponding second trial of the next session, relative to performance in the first trial of session-1." 

And we added, to the results section, the following statements: "The offline, between sessions, differences in TUG duration were -1.5 ± 3.8 percent for the interval between session-1 and session-2.” And “The offline differences in TUG duration from the end of session-2 to the beginning of session-3 were on average -4.0 ± 4.4 percent."

The corresponding percentages for the video data are now reported in the supplementary materials (Practice effects were retained across sessions (video)): 

"The offline, between sessions, differences in TUG duration were -1.3 ± 4.8, percent for the interval between session-1 and session-2." And "The offline differences in TUG duration from the end of session-2 to the beginning of session-3 were -1.6 ± 4.4, percent."

Figure:

• I agree with the authors that the trial by trial performance should be reported. However, I think that including additional bar charts showing (normalized) on- and offline learning may improve the results.

Response – Given the reviewer’s comment, we added a bar graph figure as additional supplementary figure to show the online and offline performance changes according to the videos and sensors throughout the first and last 4 trials segment of each session. 

Fig S2: Performance changes within and between segments/sessions. A) According to the videos; B) according to the IMUs. */**/***- p<0.05, p<0.01, p<0.001(repeated measures ANOVA's.

• If the data of the transfer test are not reported in the paper I would suggest to remove them from figure 2.

Response – We think that it is necessary to represent the transfer test interval in figure 2 in order to maintain the correspondence to figure 1B (the overall plan). 

Reviewer #2:

The authors comprehensively addressed all my concerns and I am happy to endorse the publication of this interesting and important article

Response – We thank the reviewer.

---

## [Decision Letter · Decision Letter 2]

25 Apr 2023

Improving old tricks as new: young adults learn from repeating everyday activities

PONE-D-22-15151R2

Dear Dr. Leizerowitz,

We’re pleased to inform you that your manuscript has been judged scientifically suitable for publication and will be formally accepted for publication once it meets all outstanding technical requirements.

Kind regards,

Peter Andreas Federolf

Academic Editor

PLOS ONE

Additional Editor Comments (optional):

Reviewers' comments:

Reviewer's Responses to Questions

**Comments to the Author**

1. If the authors have adequately addressed your comments raised in a previous round of review and you feel that this manuscript is now acceptable for publication, you may indicate that here to bypass the “Comments to the Author” section, enter your conflict of interest statement in the “Confidential to Editor” section, and submit your "Accept" recommendation.

Reviewer #1: All comments have been addressed

2. Is the manuscript technically sound, and do the data support the conclusions?

Reviewer #1: Yes

3. Has the statistical analysis been performed appropriately and rigorously? 

Reviewer #1: Yes

4. Have the authors made all data underlying the findings in their manuscript fully available?

Reviewer #1: Yes

5. Is the manuscript presented in an intelligible fashion and written in standard English?

Reviewer #1: Yes

6. Review Comments to the Author

Reviewer #1: The authors made considerable efforts in revising their manuscript and adequately addressed all my comments. Congratulations on this great work!

7. PLOS authors have the option to publish the peer review history of their article (what does this mean?). If published, this will include your full peer review and any attached files.

Reviewer #1: No

---

## [Editor Report · Acceptance letter]

3 May 2023

PONE-D-22-15151R2 

Improving old tricks as new: young adults learn from repeating everyday activities 

Dear Dr. Leizerowitz:

I'm pleased to inform you that your manuscript has been deemed suitable for publication in PLOS ONE. Congratulations! Your manuscript is now with our production department. 

Kind regards, 

on behalf of

Dr. Peter Andreas Federolf 

Academic Editor

PLOS ONE